# Divergent aging of nulliparous and parous mammary glands reveals IL33+ hybrid epithelial cells

Andrew Olander[1], Paloma Medina[1,2,3], Veronica Haro Acosta[1], Sara Kaushik[1], Matijs Dijkgraaf[1] & Shaheen S. Sikandar [1,3,4] ✉

Aging increases breast cancer risk while an early first pregnancy reduces a woman's life-long risk. Several studies have explored the effect of either aging or pregnancy on mammary stem/progenitor cells, however, the combined effect of both remains unclear. Here, we interrogate the functional and transcriptomic changes at single-cell resolution in the mammary gland of aged nulliparous and parous mice to discover that pregnancy normalizes age-related imbalances in lineage composition, while also inducing a differentiated cell state. Importantly, we uncover a minority population of *Il33*-expressing epithelial cells that express both luminal and basal markers (i.e. hybrid), which accumulate in aged nulliparous mice but are significantly reduced in aged parous mice. Functionally, IL33 treatment of mammary epithelial cells from young mice phenocopies aged nulliparous epithelial cells, induces proliferation and promotes formation of organoids with *Trp53* knockdown. Collectively, our study demonstrates that pregnancy blocks the age-associated imbalances in lineage integrity in the basal layer, including a decrease in *Il33*+ hybrid cells, that could potentially contribute to pregnancy-induced breast cancer protection.

A woman's risk of breast cancer increases with age (median age at diagnosis = 62 years[1]), but an early first pregnancy (below the age of 30) significantly reduces lifetime risk[2–8]. Previous studies that have examined changes shortly after pregnancy suggest that post-pregnancy mammary epithelial cells (MECs) have reduced stem cell capacity[9,10] and increased expression of differentiation markers[11,12]. On the other hand, the studies that examined changes with aging in nulliparous (no pregnancy) MECs found an accumulation of dysfunctional luminal progenitors[13], lineage infidelity[14], and altered differentiation programs[15,16]. Importantly, there is no clear understanding of how these processes combine and evolve with aging, i.e., how does pregnancy alter the process of aging in MECs?

Most of the previous studies focused on time points immediately post-pregnancy or 40 days post-involution (-3.3 human years),

corresponding to a transient period when the mammary gland is at a higher risk of breast cancer[17]. However, pregnancy-induced protection does not take effect until almost 10 years post-birth[4,17,18]. To address this gap in knowledge, we simulated conditions that mimic an early first pregnancy (20–30 years in humans, 3–8 months in mice) and analyzed mice at the post-menopausal stage (>50 years in humans, -18 months in mice). The long-term effect of pregnancy on the aging of MECs is important to determine because 75% of breast cancer diagnoses occur over the age of 50[19], while most women in the United States have their first pregnancy between 20 and 33 years of age[20,21]. Using this 18-month time point in conjunction with 18-month nulliparous and 3-month (nulliparous and parous) mice, we interrogate the aging process of MECs with and without pregnancy while removing confounding factors affecting human samples[22]. Our study not only

[1]Department of Molecular, Cell and Developmental Biology, University of California, Santa Cruz, CA, USA. [2]Department of Biomolecular Engineering, University of California, Santa Cruz, CA, USA. [3]Institute for the Biology of Stem Cells, University of California, Santa Cruz, CA, USA. [4]Genomics Institute, University of California, Santa Cruz, CA, USA. ✉e-mail: ssikanda@ucsc.edu

delineates the combined effects of aging and pregnancy but also uncovers a previously unknown *Il33*+ hybrid MEC population that accumulates with age and is reduced in mice that have undergone pregnancy.

## Results

### Parity induces lasting changes in the mammary gland with age

To understand whether pregnancy alters the aging of the mammary gland, we compared 18-month-old parous mice (18 M P) that have undergone multiple pregnancies between 3 and 8 months of age with 3–4 month-old nulliparous (3 M NP), 3–4 month-old parous that had undergone at least 2 pregnancies, are >3 weeks post-involution (3 M P) and 18-month-old nulliparous (18 M NP) (Fig. 1a). Hematoxylin & Eosin (H&E) staining and whole mount imaging showed no major differences in ductal density, morphology, or complexity of branching between 18 M NP and 18 M P (Supplementary Fig. 1a–d), indicating that parity does not confer long-lasting morphological changes to the mammary gland.

The mammary gland is a bilayered tree-like structure composed of two main epithelial cell lineages, luminal and basal cells[23]. Luminal

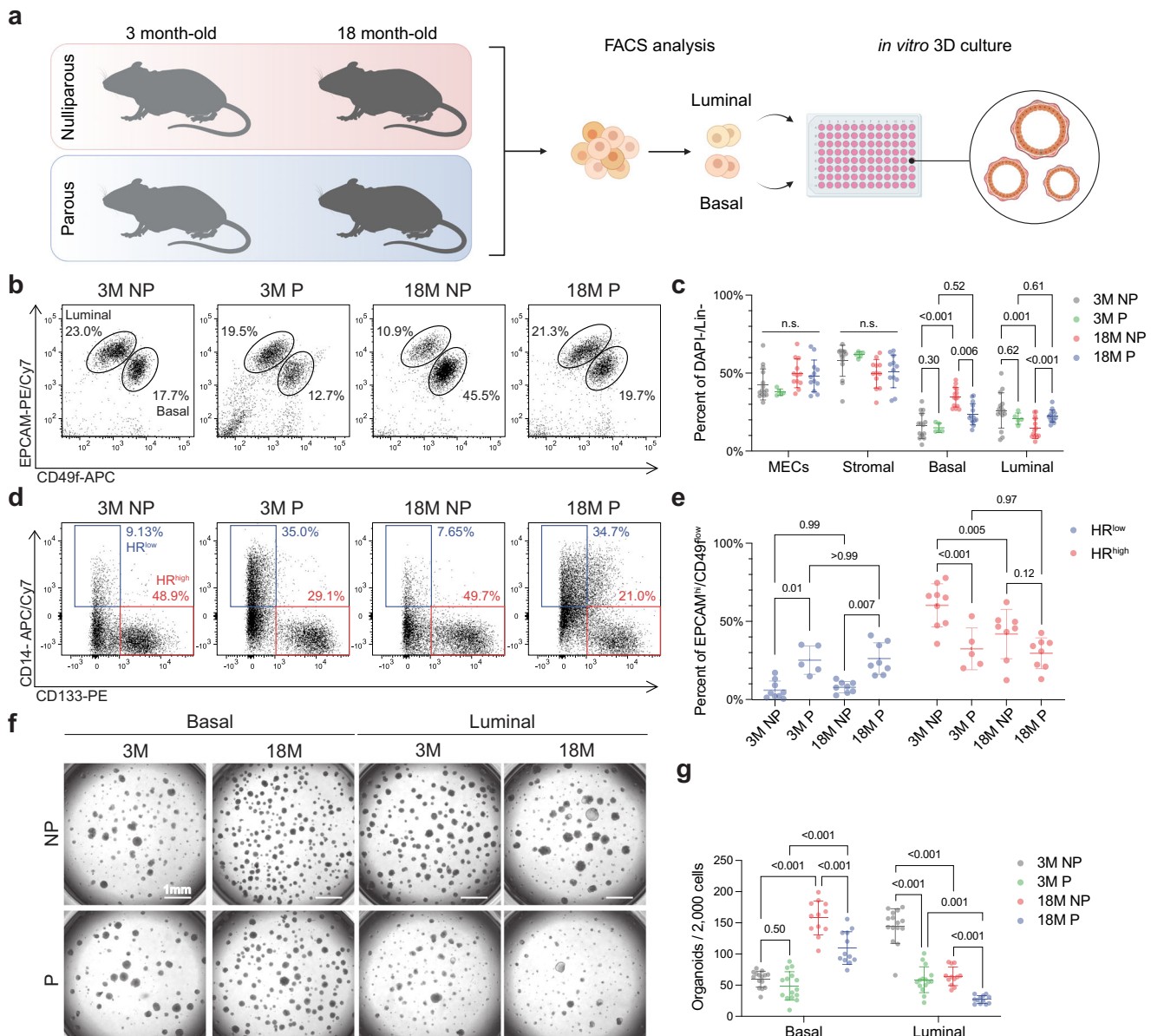

**Fig. 1 | Pregnancy induces persistent changes in cell fate decisions and reduces organoid formation. a** Schematic of flow cytometry analysis and 3D organoid cultures for 3-month nulliparous (3 M NP), 3-month parous (3 M P), 18-month nulliparous (18 M NP), and 18-month parous (18 M P) mammary glands. Created in BioRender. Olander, A. (2026) https://BioRender.com/q9uqee0. **b** Representative flow cytometry plots of DAPI-/CD45-/CD31-/Ter119- single cells from mouse mammary tissues. Basal (CD49f^high/EPCAM^med/low) and luminal (CD49f^med/low/EPCAM^high) cells are denoted by representative gates. **c** Quantification of flow cytometry data shown in (**b**). n = 15 mice/group (3 M NP); n = 5 mice/group (3 M P); n = 14 mice/group (18 M NP, 18 M P). **d** Representative flow cytometry plots of luminal cells from mouse mammary tissues. **e** Quantification of HR^low (CD14 +/

CD133- ) and HR^high (CD14-/CD133+) luminal cells shown in (**d**). n = 9 mice/group (3 M NP); n = 5 mice/group (3 M P); n = 8 mice/group (18 M NP, 18 M P). **f** Representative images of primary organoids derived from 3 M NP, 3 M P, 18 M NP, and 18 M P sorted basal (left) or luminal (right) cells. **g** Quantification of organoid colonies per well. n = 5 mice (3 M NP, 3 M P) or 4 mice (18 M NP, 18 M P) with 3 technical replicates per mouse. Statistical significance was determined by performing mixed-effects models with the Geisser-Greenhouse correction and Tukey's multiple comparisons test (**c**, **e**) or two-way ANOVA with Tukey's multiple comparisons test (**g**). Data are presented as mean values +/- S.D. (**c**, **e**, **g**). n.s. nonsignificant. Scale bar reflects 1000 μm. Source data are provided as a Source data file.

cells can be further subdivided into two functionally distinct cell types: hormone receptor (HR)-high and -low cells[24]. To understand how these populations evolve with aging in parous mammary glands, we performed flow cytometry analysis using previously established markers of MEC lineages, basal (CD49f[high]/EPCAM[med/low]) and luminal (CD49f[med]/EPCAM[high]) (Fig. 1a, Supplementary Fig. 2). We found no differences in the overall epithelial or stromal compartments among the groups (Fig. 1b, c). In 18 M NP mice, there was a significant increase in the basal population (% of lineage-/live cells) as compared to 3 M NP mice (Fig. 1b, c), consistent with previous studies[25]. However, aged parous mice showed a decrease in the basal population as compared to the aged nulliparous, suggesting that pregnancy partially normalizes the aged-induced expansion of basal cells (Fig. 1b, c). Nevertheless, there is still an age-related increase in the basal population in mice that have undergone pregnancy (3 M P versus 18 M P), although it does not reach statistical significance (Fig. 1b, c).

To further interrogate stem/progenitor subpopulations within the mammary epithelium, we assessed expression of TSPAN8, which has been reported to mark a quiescent, hormone-responsive mammary stem cell population[26]. Our analysis revealed that TSPAN8+ MECs were more abundant in young parous glands compared to age-matched nulliparous glands (Supplementary Fig. 3), suggesting an early expansion of this population following pregnancy. Interestingly, aged nulliparous glands harbored higher levels of TSPAN8+ cells relative to aged parous glands (Supplementary Fig. 3), indicating that parity may restrict the age-associated accumulation of TSPAN8+ hormone-responsive stem-like population.

Similarly, within the luminal compartment, we observed a decrease in the luminal cells with aging (3 M NP versus 18 M NP), which is normalized by pregnancy (3 M NP versus 18 M P) (Fig. 1b, c). Moreover, the percentage of luminal cells does not change with age in mice that have undergone pregnancy (3 M P versus 18 M P) (Fig. 1b, c). Our analysis of the luminal sub-lineages showed that nulliparous mammary glands have a decreased population of HR-high (CD14-/CD133+) luminal cells as a consequence of age (3 M NP vs. 18 M NP, Fig. 1d, e), consistent with previous studies in 13–14 month mice[24]. In contrast, the HR-high population was not affected by age in parous mammary glands (3 M P vs. 18 M P, Fig. 1d, e). However, regardless of age, parity resulted in a robust reduction in the HR-high population, suggesting that the pregnancy-driven reduction in HR-high cells is maintained with age. Alternatively, this reduction of the HR-high luminal population in parous mice may represent a consequence of the expanded HR-low (CD14+/CD133-) luminal population, which was exclusive to parous mammary glands and unaffected by age (Fig. 1d, e). Given the observed upregulation of CD14 in luminal cells during involution[27], our data suggest that while the age-induced expansion of basal cells is normalized by pregnancy, luminal cells retain a residual involution program, with an increased proportion of CD14 + HR-low luminal cells.

Terminal differentiation of stem and progenitor cells during pregnancy has been hypothesized to contribute to pregnancy-induced breast cancer protection[6,8,9]. To understand whether there are functional differences in the clonogenicity of basal or luminal cells between the 3 M NP, 3 M P, 18 M NP, and 18 M P mice, we performed in vitro organoid assays. In line with previous studies that have shown an age-induced increase in organoid formation[25], we found that aged basal cells had the highest capacity to form organoids as compared to young basal cells, regardless of parity status (Fig. 1f, g). While basal cells from aged parous mice had a higher organoid formation capacity than young mice (3 M P and NP vs. 18 M P), the number of organoids was significantly lower compared to aged nulliparous mice (18 M NP versus 18 M P) (Fig. 1f, g). These data suggest that pregnancy normalizes some of the age-induced increase in the clonogenicity of basal cells. Luminal cells from 3 M P mice had significantly reduced organoid formation capacity as compared to 3 M NP (Fig. 1f, g). Interestingly, aging reduced organoid-forming capacity of luminal cells, and this was further reduced in aged parous mice (lowest organoid formation capacity) (Fig. 1f, g). These findings suggest that MECs from the aged parous glands are diminished in clonogenicity, supporting the hypothesis that pregnancy reduces regenerative potential.

## Single-cell RNA-seq reveals rare hybrid MECs in aged nulliparous mice

Recent studies have demonstrated that minority populations of stem/progenitor cells evolve with age and are likely precursors of tumor initiation[25,28]. To investigate the cellular heterogeneity and aging-associated transcriptional changes that are altered with pregnancy at single-cell resolution, we performed single-cell RNA-sequencing (scRNA-seq) on mammary glands from 18 M NP and 18 M P mice. To increase the power of our analysis, we integrated single-cell transcriptomes from the Tabula Muris[29] 18 M mammary gland dataset (see Methods). After filtering cells with low and high-expressing genes, high mitochondrial counts, and doublets (Supplementary Fig. 4, see Methods), we identified epithelial cells (basal, HR-high luminal, HR-low luminal), immune cells (B cells, T cells, dendritic, macrophages), endothelial cells, pericytes, and fibroblasts using the expression of well-established marker genes (Fig. 2a, b, Supplementary Fig. 5a). Intriguingly, we identified a minority population of cells co-expressing basal and luminal markers (hybrid) (Fig. 2c), that are present in 18 M NP mice (~97% of cell cluster) but are significantly reduced in 18 M P mice (~3% of cell cluster) (Fig. 2b). Moreover, we found no major differences in the proportion of B cells, T cells, macrophages, endothelial cells, dendritic cells, pericytes, fibroblasts, and HR-low luminal cells between 18 M P and 18 M NP (Fig. 2b). However, 18 M P had a lower proportion of basal cells and HR-high luminal cells (Fig. 2b), consistent with our flow cytometry analyses (Fig. 1b−e). Moreover, we did not find differences in cell cycle between 18 M NP and 18 M P mice (Supplementary Fig. 5b).

To investigate the unknown hybrid population further, we calculated basal cell and luminal cell gene signature enrichment scores for all cells based on previously published transcriptomic analyses of adult MECs[30]. We found that the hybrid MEC population lies between the luminal and basal populations, implying that hybrid MECs display a mixed gene signature of both basal and luminal lineages (Fig. 2d). Pseudotime inference using PAGA[31] and Slingshot[32] also predicted the hybrid MECs lineage trajectory between basal and luminal populations (Fig. 2e, Supplementary Fig. 6a). To further understand whether these hybrid MECs represent a plastic/intermediate cell state, we applied CytoTRACE2[33] to agnostically predict cellular potency. The average cellular potency of hybrid MECs scored between basal and luminal cells (Fig. 2f), supporting our hypothesis that these cells represent a hybrid intermediate.

Interestingly, we found that the hybrid MECs are enriched in expression of *Krt6a* and *Il33* (Fig. 2c, Supplementary Fig. 6b). KRT6A has been previously identified as a marker of bipotent luminal progenitor cells in the mammary gland[34]. KRT6A+ cells are normally found only in the luminal layer during development[35] and do not express the basal marker *Krt14*. However, a previous report showed that a population of KRT14+/KRT6A+ cells expands during early stages of pregnancy[35]. IL33 has been previously implicated in activating pro-tumorigenic signaling pathways[36–38], cancer stem cell maintenance[39], and establishing an immunosuppressive microenvironment[40]. Although other genes such as *Aldh3a1, Dmkn, Alcam, Ly6d, Dsc3, Ppp1r14c, S100a14, Lgals7, Dapl1, Atp1b1, Cldn10,* and *Anxa8* are enriched in the hybrid population (Fig. 2c), their expression is either not exclusive to this population or limited to a smaller fraction of cells in the cluster. We therefore used *Krt6a* and *Il33* as markers to further understand the hybrid population.

## *Il33*+/*Krt6a*+ cells are present in public datasets

To validate our findings in other published aging datasets, we used the recently published data from 3 M and 18 M NP datasets from Angarola

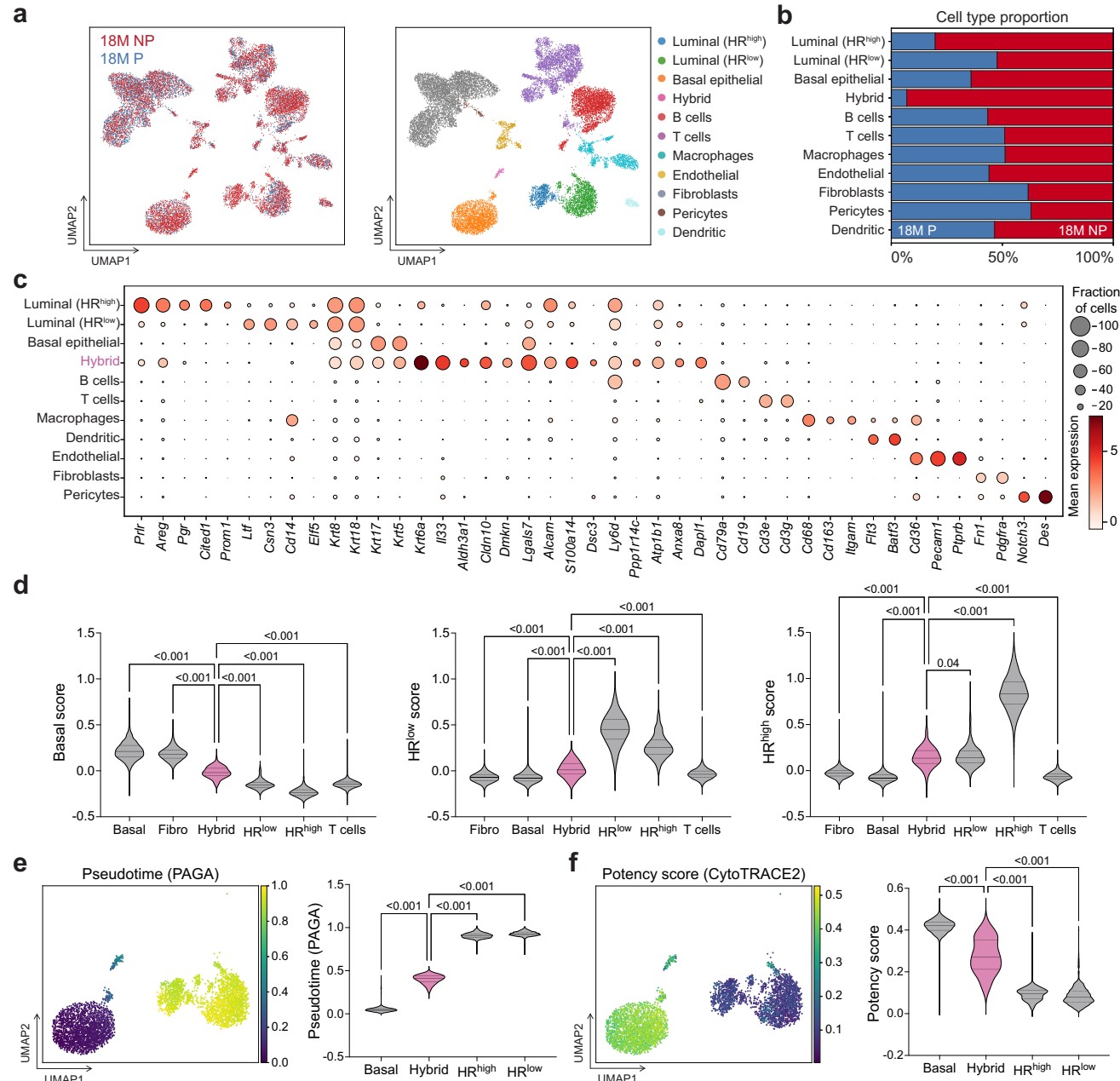

**Fig. 2 | Single-cell RNA sequencing identifies an age-dependent accumulation of hybrid MECs in nulliparous mice. a** UMAP plot of scRNA-seq data generated from the mammary glands of 18-month nulliparous (NP, red, 8131 cells, 2058 epithelial cells) and parous (P, blue, 7305 cells, 1336 epithelial cells) mice ($n = 3$ mice per group) and integrated with Tabula Muris (red, $n = 2993$ cells, 1311 epithelial cells). UMAPs are colored by condition (left) and annotated cell type (right). **b** Bar plot showing the proportion of cell types by condition. Data were down-sampled to reflect equal numbers of parous and nulliparous cells. **c** Dotplot of gene expression for selected marker genes of cell lineages in the mammary gland. **d** Basal, HR-low, and HR-high gene signature scores across epithelial clusters (gray), including a minority population of hybrid cells (pink). Fibroblasts and T cells (gray) are included as controls. **e** UMAP and violin plot of inferred partition-based graph abstraction (PAGA) pseudotime scores across epithelial cell clusters. **f** UMAP and violin plot comparing predicted developmental potential (obtained through CytoTRACE 2) across epithelial cell clusters. Statistical significance was determined by performing a one-way ANOVA with Holm-Šídák's multiple comparisons test, and $p$-values are displayed for comparisons between hybrid cells and other populations only (**d–f**). Data are presented as violin plots with quartiles represented by dashed lines and median values represented by solid lines (**d–f**). Source data are provided as a Source data file.

et al.[41] and the Smart-Seq2 data from Tabula Muris Senis (note: independent from the 10x dataset used for integration). Consistent with our dataset, we identified a minority *Krt6a+/Il33+* population that appears as a distinct cluster (shown in pink) and was labeled as myoepithelial in the Angarola et al. dataset (Fig. 3a–c, 39 cells in the cluster) and basal in the Smart-Seq data (Fig. 3d–f, 103 cells in the cluster), although a small percentage of luminal cells are also *Krt6a+/Il33+* (Fig. 3f). The proportion of *Krt6a+/Il33+* cells is highest in the

aged mammary glands as compared to young mammary glands, further confirming that an increase in the *Krt6a+/Il33+* population occurs with aging (Fig. 3c, f).

To understand the dynamics of the hybrid population with pregnancy and aging, we integrated single-cell RNA-sequencing data from the mammary glands of 3-month-old nulliparous and 3-weeks post-involution mice from the Pal et al.[42] dataset with our data. The highest proportion of *Krt6a+/Il33+* cells were found in the 18 M NP

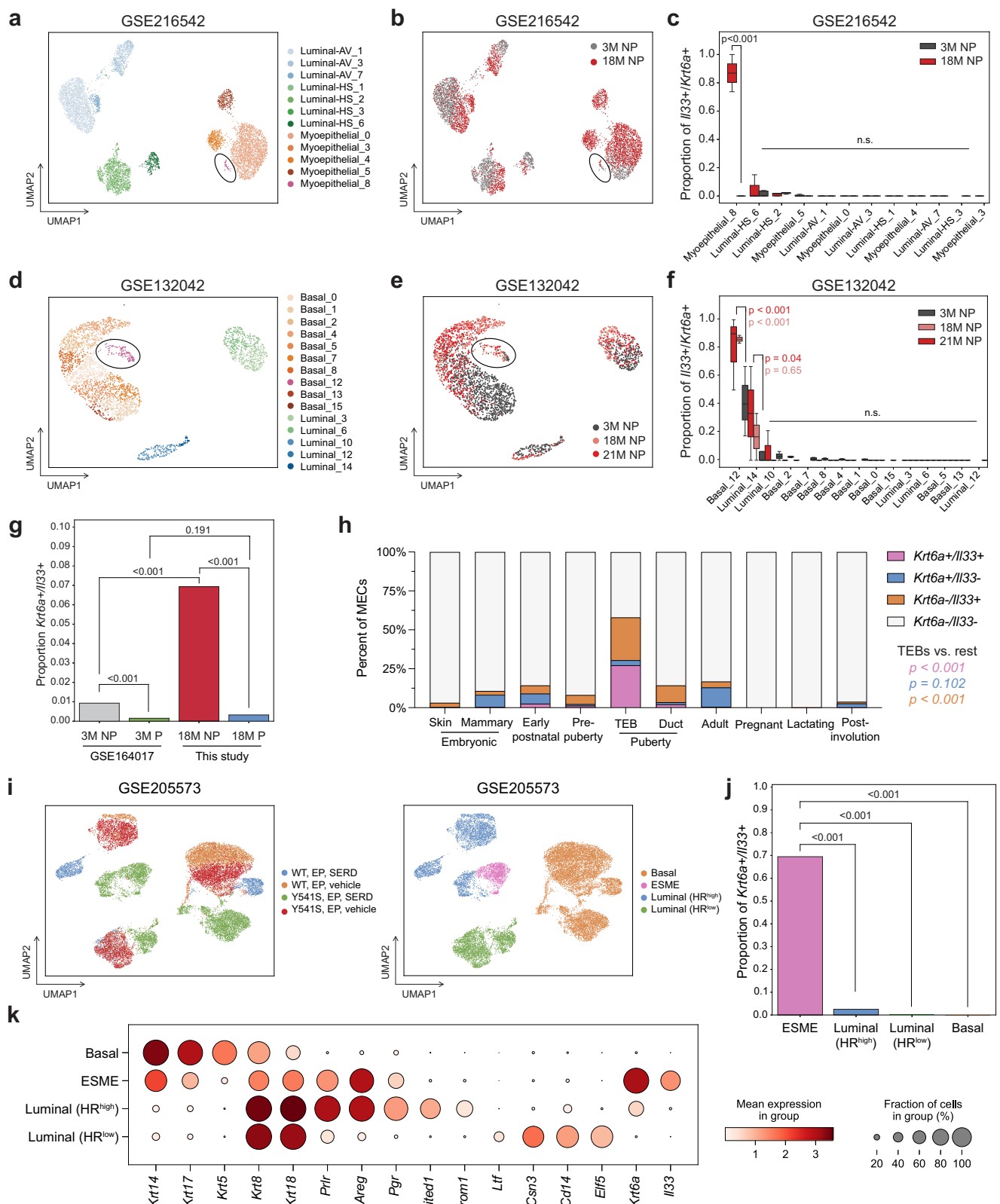

samples (~7%), with a much smaller proportion found in both 3 M NP and 18 M P samples (<1% in both, Fig. 3g). Intriguingly, there is a decrease in *Krt6a+/Il33+* cells post-pregnancy (Fig. 3g, <0.3%), suggesting that the change in the proportion of cells occurs immediately after pregnancy but the accumulation in nulliparous mice is an age-dependent effect.

To understand whether the *Krt6a+/Il33+* population is present during the different stages of mammary gland development, we analyzed the Pal et al.[42] scRNA-seq dataset containing all mammary gland developmental stages. We found that *Krt6a* and *Il33* expression peaks during early pubertal development, particularly within terminal end buds (TEBs)[42] (Fig. 3h, Supplementary Fig. 6c). Taken together, these

**Fig. 3 | *Krt6a+/Il33+* hybrid MECs are enriched in TEBs and accumulate in aged and *Esr1* mutant mammary glands. a, b** UMAP plot of scRNA-seq data generated from the mammary glands of 3-month nulliparous (3 M NP, gray) and 18-month nulliparous (18 M NP, red) mice (GSE216542). UMAPs are colored by annotated cell type subclusters (**a**) and age group (**b**). **c** Proportion of cells expressing *Il33* and *Krt6a* in each subcluster, split by age group. Whiskers reflect the minimum and maximum values, the center line reflects the median, and the box represents the interquartile range (minimum-median-maximum values for Myoepithelial_8 across ages from left to right are as follows: 0.73684211-0.86842105-1, 0-0-0), 5 mice per condition. **d, e** UMAP plot of scRNA-seq data generated from the mammary glands of 3-month nulliparous (3 M NP, gray), 18-month nulliparous (18 M NP, pink), and 21-month nulliparous (21 M NP, red) mice (GSE132042). UMAPs are colored by annotated cell type subclusters (**d**) and age group (**e**). **f** Proportion of cells expressing *Il33* and *Krt6a* in each sub-cluster, split by age group. Whiskers reflect the minimum and maximum values, center line reflects the median, and the box represents the interquartile range (minimum-median-maximum values for Basal_12 across ages from left to right are as follows: 0.5- 0.9-1, 0.83333333-0.86111111-0.88888889, 0.17391304-0.4-0.66666667), 3 mice per condition. **g** Proportion of cells expressing *Il33* and *Krt6a* across 3 M NP, 3 M P, 18 M NP, and 18 M P datasets. **h** Percentage of cells expressing *Il33* and/or *Krt6a* or neither across mammary gland developmental stages (GSE164017). **i** UMAP plots of scRNA-seq data from wildtype (WT) or *Esr1* mutant (Y541S) mammary glands treated with E2/P4 (EP) and ER-alpha antagonists and degraders (SERD) or vehicle control. Clusters are colored according to condition (left) and annotated cell type (right). **j** Proportion of cells expressing *Il33* and *Krt6a* across cell types. **k** Dotplot depicting gene expression for epithelial lineage markers and hybrid cell markers across cell types. Statistical significance was determined by performing one-way ANOVA with Tukey's multiple comparisons test (**c, f**) or two-sided Chi square tests (**g, h, j**). *p*-values are displayed for comparisons between hybrid cells and other populations only (**h, j**). n.s. non-significant. Source data are provided as a Source data file.

data suggest that aging coincides with an aberrant activation of *Krt6a* and *Il33* expression, likely conferring a cell state similar to TEBs, but this is diminished by pregnancy.

An early pregnancy primarily protects women from ER+ breast cancer[43]; hence, we wanted to understand the relevance of our hybrid population found in aged nulliparous mice to this disease in the pre-malignant phase. Therefore, we used the recently published single-cell dataset generated from mouse mammary glands carrying mutations in *Esr1* that arise after developing resistance to anti-estrogen therapies[44] (Fig. 3i). Remarkably, we find that the cells that emerge after estrogen deprivation or inhibition in mutant *Esr1* mammary glands, labeled as endocrine-suppressed mutant-enriched (ESME), are highly enriched in the proportion of *Krt6a + /Il33+* cells (Fig. 3j) and expression of *Krt6a* and *Il33* (Fig. 3k). These findings suggest that aging results in the emergence of a hybrid cell population that shares features of *Esr1* mutant cells with high cellular plasticity.

### KRT6A+ cells are present in aged nulliparous but not parous mammary glands

To expand our analysis from mRNA to protein, we performed immunofluorescence staining on 3 M NP, 3 M P, 18 M NP, and 18 M P mammary glands for KRT6A, together with KRT8 (luminal marker) and KRT5 (basal marker) (Fig. 4a). We found a 4.5-fold increase in the percentage of KRT6A+ cells in the 18 M NP (6.1%) mice relative to 3 M NP (1.34%) (Fig. 4b), suggesting that these KRT6A+ MECs accumulate over the course of aging. However, 18 M P mice had a small percentage of KRT6A+ cells (<1%), less than the 3 M NP (Fig. 4b), consistent with our single-cell analysis (Fig. 3g). We could not detect more than a couple of KRT6A+ cells in 3 M P mice, suggesting that there is a decrease in the population at 3 weeks post-involution (Fig. 4b), again consistent with our single-cell analysis (Fig. 3g).

To further understand the localization of the KRT6A+ cells, we quantified the percentage of KRT6A+ cells in the luminal and basal layer (i.e., KRT6A + /KRT8+ or KRT6A+/KRT5+). We found that 95% of KRT6A+ cells in 3 M NP glands were positive for KRT8 and resided in the luminal layer (Fig. 4b), consistent with previous studies[34]. In contrast, over 50% of KRT6A+ cells in the 18 M NP gland were positive for KRT5 and localized to the basal layer. Interestingly, in the 18 M P mice, the localization of the KRT6A+ cells was similar to 3 M NP mice (~10%, Fig. 4b). This suggests that the KRT6A+ MECs that accumulate with age are likely due to the loss of lineage integrity of basal cells, but this integrity is maintained in mice that have undergone pregnancy. Moreover, we found an increased percentage of KRT5+ basal cells and a decreased percentage of KRT8+ luminal cells in 18 M NP mice (Fig. 4c), consistent with our flow cytometry data (Fig. 1b, d), but 18 M P cells were more similar to 3 M NP cells.

To determine whether the phenotype is maintained ex vivo, we performed immunofluorescence staining on 3 M NP, 3M P, 18 M NP, and 18 M P organoids derived from basal or luminal cells (hereafter referred to as basal organoids or luminal organoids) for KRT6A, KRT8, and KRT5. Consistent with the staining in the mammary gland, we found that basal organoids from 18 M NP mice had the highest percentage of KRT6A+ cells (Fig. 4d, e). Moreover, we observed very few KRT6A+ cells in organoids derived from luminal cells (Fig. 4f, g). Flow cytometry analysis of the organoids using markers in Fig. 1 showed trends in basal and luminal cell proportions comparable to immunofluorescence staining experiments (Supplementary Fig. 7). Of note, the proportions of epithelial populations appear to be mostly dependent on age in basal organoids, but parity status in luminal organoids. These data suggest that basal cells from 18 M NP mice retain the aberrant differentiation program in short-term in vitro culture.

### In vitro IL33 treatment mimics aged nulliparous profile

Cellular plasticity and hybrid cell states have been proposed as permissive cell states that allow for malignant transformation. Although IL33 has been studied in the context of breast cancer[36,39], its role in the normal mammary gland, cellular plasticity, and aging is not defined. To determine whether IL33 treatment affects primary MECs in vitro, we cultured basal or luminal cells from 3-month old nulliparous (3 M NP) mice with recombinant IL33 (5 or 10 ng/mL). The treatment of basal and luminal cells with IL33 resulted in an increased number of basal-derived organoids (Fig. 5a) but had no effect on organoids derived from luminal cells (Fig. 5b). Interestingly, IL33 treatment of basal organoids for 2 weeks inhibited spontaneous in vitro luminal cell differentiation, resulting in an increase in the proportion of basal cells (Fig. 5c, d) but no differences were observed in luminal cells (Fig. 5e). Moreover, immunofluorescence staining and 3D imaging of organoids demonstrated an increase in the percentage of KRT6A+ and KRT5+ cells upon treatment with IL33 in basal cells, and a slight but non-significant decrease in KRT8+ cells (Fig. 5f, g). These changes were not observed in organoids derived from luminal cells (Fig. 5g, bottom panel), suggesting that IL33 treatment of young nulliparous basal cells, but not luminal cells, phenocopies the increased proportion of basal cells and accumulation of KRT6A+ hybrid MECs observed in aged nulliparous mice (Fig. 4a, b).

To test whether IL33 treatment increases clonogenicity in cells with tumor-promoting mutations, we transduced 3 M NP basal and luminal cells with lentiviral shRNA targeting *Trp53*[45] (Supplementary Fig. 8) and exposed them to IL33. We found that *Trp53* knockdown resulted in increased clonogenicity of basal cells when treated with IL33, but not luminal cells (Fig. 5h). Thus, our data suggest that the age-induced increase in *Il33*+ hybrid MECs could confer a survival advantage as cells acquire oncogenic mutations.

### In vivo IL33 treatment drives proliferation and alters lineage identity

To determine whether IL33 treatment phenocopies an age-related change in vivo, we treated young nulliparous mice (3 M NP) with IL33,

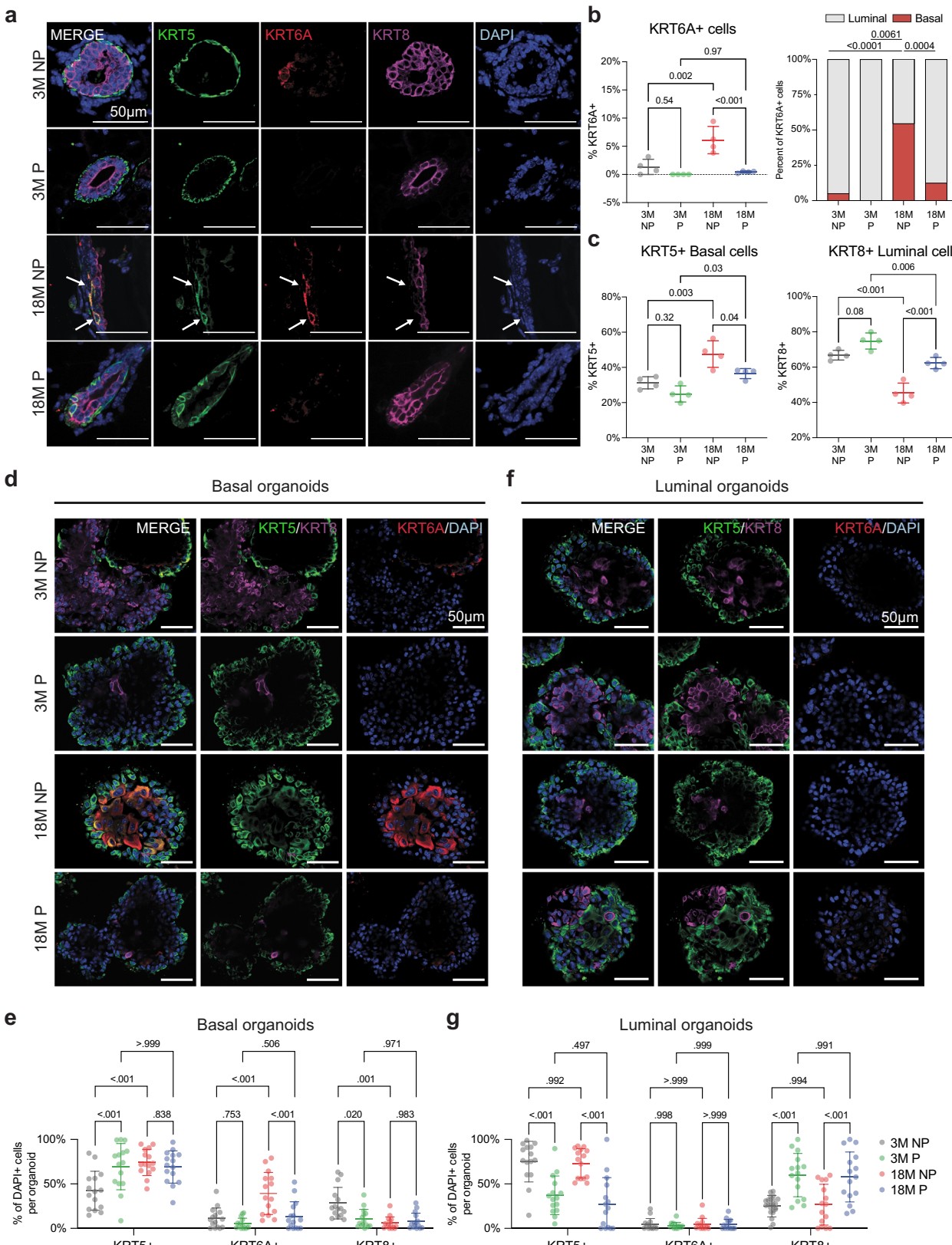

or PBS for 5 days as previously described[46] and analyzed mammary glands at 1 and 8 days post-treatment (dpt) (Fig. 6a). Wholemount analysis of the mammary glands from IL33-treated mice showed a significantly increased of the lymph node and dramatic dilation of the ducts at 1dpt (Fig. 6b, c). This phenotype closely resembles mammary glands transplanted with MECs expressing constitutively

active PIK3CA[47] and in mammary glands where FGF3 expression is ectopically induced[48]. There were no differences in the length of the primary duct (Fig. 6d). H&E staining confirmed a significant increase in size of the lumen in the IL33-treated group at 1dpt (Fig. 6e, f) but did not reveal fluid accumulation within the ducts; however, we cannot entirely exclude the possibility of dilation secondary to edema. Aside

**Fig. 4 | KRT6A+ cells are reduced by pregnancy and are in the basal layer.**
**a** Representative immunofluorescence images for KRT5 (basal cells, green), KRT6A (red), and KRT8 (luminal cells, magenta). Nuclei are visualized using DAPI (blue). KRT6A+ cells localized to the basal layer are indicated by white arrows.
**b** Quantification of total KRT6A+ cells in the epithelium (left) and localization of KRT6A+ cells (right). $n = 4$ mice per group, data points represent the average from 4 sections per mouse. **c** Quantification of total KRT5+ basal cells (left) and KRT8+ luminal cells (right) in the mammary epithelium. $n = 4$ mice per group, data points represent average from 4 sections per mouse. Representative immunofluorescence images in basal organoids (**d**) and luminal organoids (**e**) from 3-month nulliparous

(3 M NP), 3-month parous (3 M P), 18-month nulliparous (18 M NP), and 18-month parous (18 M P). Organoids were stained for KRT5 (basal cells, green), KRT6A (red), and KRT8 (luminal cells, magenta). Nuclei are visualized using DAPI (blue). Quantification of KRT5+ basal cells, KRT6A+ cells, and KRT8+ luminal cells in basal (**f**) and luminal (**g**) organoids across comparison groups ($n = 15$ organoids derived from 3 mice per group). Statistical significance was determined by performing one-way ANOVA with Tukey's multiple comparisons test (**b**, **c**) and two-way ANOVA with Tukey's multiple comparisons test (**f**, **g**) or two-sided Chi square test (**b**, right). Data are presented as mean values +/- S.D. (**b**, **c**, **e**, **g**). Scale bar reflects 50 μm. Source data are provided as a Source data file.

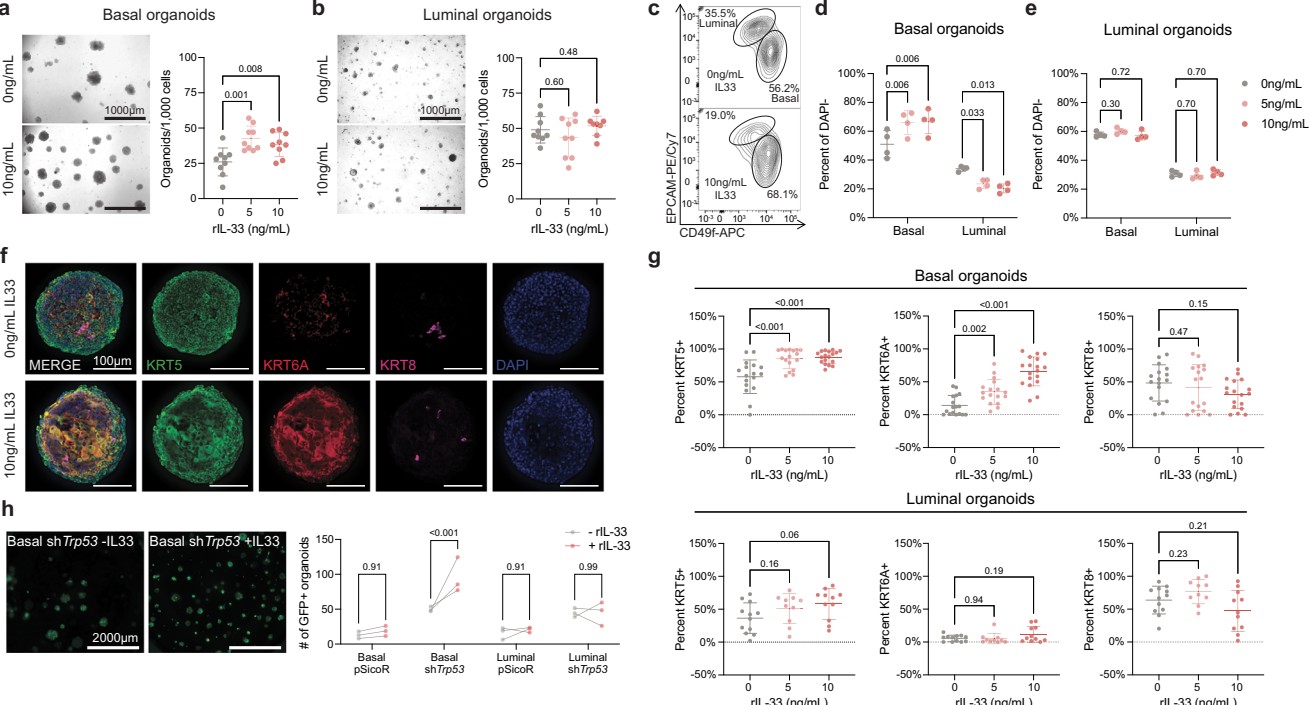

**Fig. 5 | IL33 treatment of young basal cells phenocopies aged nulliparous MECs.**
**a** Representative images of primary basal organoids from 3 M NP mice with no IL33 treatment (top left) and with IL33 treatment (bottom left). Quantification of the number of organoids formed per 1000 cells seeded with 0, 5, or 10 ng/mL of IL33. $n = 9$ individual wells for each condition from 3 mice. **b** Representative images of primary luminal organoids from 3 M NP mice with no IL33 treatment (top left) and with IL33 treatment (bottom left). Quantification of the number of organoids formed per 1000 cells seeded with 0, 5, or 10 ng/mL of IL33. $n = 9$ (0 ng/mL, 5 ng/mL) or 8 (10 ng/mL) wells for each condition from 3 mice. **c** Representative flow cytometry plots of DAPI- single cells from untreated (top) and IL33-treated (bottom) basal organoids. Basal and luminal cells are denoted by representative gates. Percentage of basal and luminal cells as a fraction of DAPI- single cells in basal (**d**) and luminal (**e**) organoids exposed to 0, 5, or 10 ng/mL of IL33. $n = 4$ mice for each condition. **f** Representative immunofluorescence images for KRT5 (basal cells), KRT6A, and KRT8 (luminal cells) on IL33-treated basal organoids. Nuclei are

visualized using DAPI. **g** Percentage of KRT5+ basal, KRT6A+, and KRT8 luminal cells in basal (top panel) and luminal (bottom panel) organoids exposed to 0, 5, or 10 ng/mL of IL33. For basal organoids, $n = 17$ (0 ng/mL, 5 ng/mL) or 18 (10 ng/mL) organoids derived from 3 mice. For luminal organoids, $n = 12$ (0 ng/mL), 10 (5 ng/mL), or 11 (10 ng/mL) organoids derived from 2 mice. **h** Representative images of preneoplastic (sh*Tp53*) basal organoids treated with (left image, 10 ng/mL) or without IL33 (right image). Quantification of the number of pSicoR or sh*Trp53* basal and luminal organoids formed with or without IL33 treatment after 12 days in culture (right). $n = 3$ mice for each condition. **Note: h** basal organoids transduced with pSICOR became non-proliferative, dampening the response to IL33, but treated and untreated groups are significant at 0.05 (paired two-sided *t*-test). Statistical significance was determined by performing one-way (**a**, **b**, **g**) or two-way (**d**, **e**, **h**) ANOVA with Holm-Šídák's multiple comparisons test. Data are presented as mean values +/- S.D. (**a**, **b**, **d**, **e**, **g**). Source data are provided as a Source data file.

from the area of the lymph node, analysis of mice at 8dpt showed no significant changes in any of the parameters above, suggesting the effects of short-term IL33 treatment were transient (Fig. 6b–f).

To further investigate whether IL33 treatment altered the cell-type composition in the mammary gland, we performed flow cytometry analysis at 1dpt and 8dpt for epithelial and immune cell types. We find that IL33 treatment reduces the total number of MECs, largely due to an increase in the percentage of stromal cells and a decrease in the percentage of luminal cells resulting in an overall increase in basal cells (Fig. 6g). To understand whether these changes are due to proliferation or apoptosis, we performed EdU labeling during and after

treatment (Fig. 6a) as well as Annexin V staining. Analysis of mice at 1dpt showed that IL33 induces significant proliferation of both basal (~2-fold increase) and luminal cells (~4-fold increase) (Fig. 6h), while no differences were observed by 8dpt. At 1dpt and 8dpt, there are small but significant differences in apoptosis in the basal compartment (Supplementary Fig. 9). Specifically, we observed a decreased percentage of cells in early apoptosis (DAPI-/AnnexinV+) in basal cells at both 1 and 8dpt (Supplementary Fig. 9b, *left*). While insignificant at 1dpt ($p = 0.06$), we also observed a slightly elevated live population (DAPI-/AnnexinV-) in IL33-treated mice (Supplementary Fig. 9b, *left*). Within the luminal population, we find a non-significant increase in

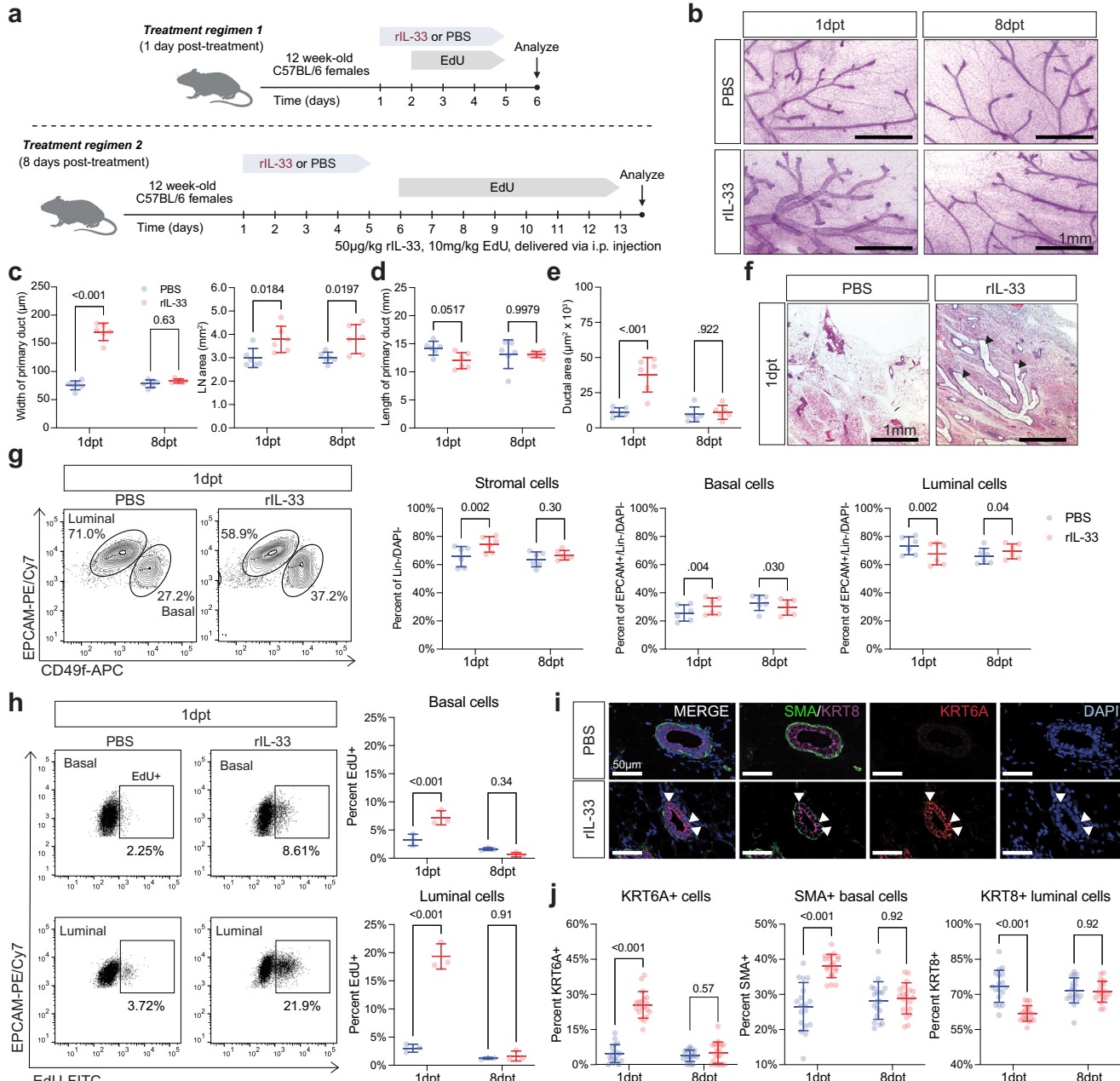

**Fig. 6 | IL33 alters epithelial differentiation in young nulliparous mice.**
**a** Experimental design to investigate the effects of rIL-33 treatment in young mammary glands. 3-month nulliparous mice were injected (intraperitoneal) with 1 μg of recombinant IL-33 (rIL-33) daily for 5 days and 10 mg/kg EdU for the indicated duration. Mice were analyzed at 1 and 8 days post treatment (dpt) for wholemount, flow cytometry, and immunofluorescence analyses. Created in BioRender. Olander, A. (2026) https://BioRender.com/mq39ceb. **b** Representative images of PBS- and rIL-33-injected mammary gland wholemounts stained with Carmine Alum. **c** Width (μm) of the main epithelial duct (n = 6 mice per condition, data points represent an average of 5 or 10 measurements per mouse) and lymph node (LN) area (n = 6 mice per condition) in PBS- and rIL-33-injected mammary glands at 1dpt and 8dpt. **d** Length (mm) of the main epithelial duct in PBS- (blue) and rIL-33-injected (red) mammary glands at 1dpt and 8dpt (n = 6 mice per condition). **e** Lumen area (μm² × 10³) in H&E stained sections of PBS- and rIL-33-injected mammary glands at 1dpt (n = 6 mice per condition, data points represent an average of 20 measurements per mouse). **f** Representative images of PBS- and rIL-33-injected H&E stains at 1dpt and 8dpt. **g** Representative flow cytometry plots of

EpCAM + /DAPI-/CD45-/CD31-/Ter119- single cells from PBS- and rIL-33-injected mammary glands at 1dpt (left). Basal and luminal cells are denoted by representative gates. Percentage of stromal cells, basal cells, and luminal cells as a fraction of Lin- population (right, n = 6 mice per condition). **h** Representative flow plots depicting EdU+ gates for basal (top panel) and luminal (bottom panel) cells across treatment groups (PBS and IL33) and timepoints (1dpt) (left). Quantification of EdU+ basal (top) and luminal (bottom) cells across treatment groups (PBS and IL33) and timepoints (1dpt and 8dpt) (right, n = 3 mice per condition).
**i** Representative immunofluorescence images for SMA (basal cells), KRT6A, and KRT8 (luminal cells) on PBS- and rIL-33-injected mammary glands at 1dpt. Nuclei are visualized using DAPI. **j** Percentage of SMA+ basal, KRT6A+, and KRT8+ luminal cells in PBS- and rIL-33-injected mammary glands at 1dpt and 8dpt. n = 18 (1dpt IL33) or 19 (1dpt PBS, 8dpt PBS, 8dpt IL33) sections across 4 mice per condition. Statistical significance was determined by performing ANOVA with Holm-Šídák's multiple comparisons test. Scale bar reflects 1 mm (**b**, **f**) or 50 μm (**i**). Data are presented as mean values +/- S.D. (**c–e**, **g**, **h**, **j**). Source data are provided as a Source data file.

apoptosis with IL33 treatment at 1dpt, which is reversed at the 8dpt (Supplementary Fig. 9b, *right*). To determine whether ductal dilation was due to an increase in cell size, we used forward scatter area (FSC-A) as a proxy for cell size. We observed a small decrease in median FSC-A in IL33-treated MECs at 1dpt (Supplementary Fig. 9c, d). However, this difference was lost when we excluded the EdU+ cells, suggesting that the ductal dilation is not primarily due to an increase in cell size (Supplementary Fig. 9e). Given the relatively minor changes in apoptosis and cell size, we speculate that the effects of IL33 are largely due to a dramatic increase in proliferation and not cell death. Moreover, we did not see significant changes in immune cell composition (Supplementary Fig. 10a, b) except an increase in ILC2 cells, the responder cell to IL33[46,49,50], confirming the response of the treatment (Supplementary Fig. 10c, d).

We next tested whether IL33 treatment induces KRT6A expression in the mammary gland. Remarkably, IL33 increased the percentage of KRT6A+ hybrid cells in mice at 1dpt but not in mice at 8dpt (Fig. 6i, j). Moreover, while IL33 treatment in vitro induces KRT6A expression in basal cells, the in vivo treatment induces expression in luminal cells. This suggests that although IL33 induces the KRT6A+ cell state, the responding cell type to exogenous IL33 is distinct in vitro and in vivo in young mice. This discrepancy could be due to the short-term in vivo treatment, lower proliferation of basal cells in vivo, and/or limited differentiation into luminal cells in vivo as compared to organoid cultures. In addition, IL33 treatment in vivo increased the percentage of SMA+ basal cells and decreased the percentage of KRT8+ luminal cells (Fig. 6i, j) consistent with our flow cytometry analysis (Fig. 6g), suggesting the IL33 treatment can partially recapitulate some age-related phenotypes in vivo (Fig. 1).

### IL33+ hybrid MECs are present in normal human breast tissue

We next sought to determine whether the *Il33+* hybrid MEC population is present in human samples. To this end, we sub-clustered epithelial populations from the Human Breast Cell Atlas (HBCA)[51] scRNA-seq dataset. We identified a population of cells (sub-cluster 5, pink) that were classified as HR[low] luminal cells, though they formed a distinct cluster separate from the major HR[low] luminal cluster (Fig. 7a). Cells in sub-cluster 5 co-expressed basal and luminal markers (Fig. 7b, Supplementary Fig. 11a), suggesting that this cluster represents an intermediate cell state between basal and luminal cells. Within sub-cluster 5, we discovered a minority population of cells enriched in the expression of genes present in our hybrid MEC population, as seen in Fig. 3c, such as *IL33, LY6D, ALCAM, DMKN, PPP1R14C*, and *DAPL1*, among others (Fig. 7c, Supplementary Fig. 11b). Interestingly, this hybrid population is enriched in expression of *MKI67* and *TOP2A* (Supplementary Fig. 11b), potentially indicating a connection between IL33 and aberrant proliferation.

However, we were unable to identify robust expression of *KRT6A* (Supplementary Fig. 11c), likely reflecting differences in mouse and human keratin markers. Moreover, expression of these hybrid genes is confined to a relatively small population of cells within the cluster; this may be due in part to technical limitations such as dropout events or low sequencing depth that specifically affects genes that are expressed at low levels[52,53]. Furthermore, we did not observe differences in age or parity within this subcluster (Supplementary Fig. 11d). We therefore explored whether the percentage of *IL33+* cells change with age or parity. We found that there is a significant increase in *IL33+* cells with age in nulliparous women but no difference between aged nulliparous and parous women (Supplementary Fig. 11e). However, since the HBCA[51] dataset does not include information on age at first pregnancy and a late pregnancy may not have the same effect, we are unable to conclusively determine whether this population declines in older women who experienced early pregnancy. Similarly, we analyzed the iHBCA[22] to test whether the percentage of *IL33+* cells changes with age (Supplementary Fig. 12a). First, we were able to identify a population of

hybrid cells that is enriched in expression of *IL33* (Supplementary Fig. 12b). Specifically, this distinct hybrid cluster expresses both basal and luminal markers (Supplementary Fig. 12b, black dashed circle) and is enriched in the Kumar et al. dataset. Moreover, in the Twigger et al. dataset[54], there is also a population of *IL33+* cells that only expresses luminal markers (Supplementary Fig. 12b, pink dashed circle). As the Twigger et al. dataset was generated using cells isolated from human milk, we speculate that the expression of *IL33* could reflect unique biology associated with lactation and immunity. Although a distinct *IL33+* cluster could not be identified in the other datasets that are part of the iHBCA, we were able to identify *IL33+* cells in all of them (Supplementary Fig. 12c). Therefore, we explored whether the percentage of *IL33+* epithelial cells and/or expression of *IL33* changes with age in the iHBCA. Overall, we find that the percentage of *IL33+* cells increases with age in all datasets except the Twigger et al. dataset (Supplementary Fig. 12d). Like the HBCA[51], we could not identify enough aged patients that had an early pregnancy and no mutations in DNA-repair pathways, precluding our analysis with early pregnancy. Despite these discrepancies between our analysis of mouse and human scRNA-seq datasets, it is apparent that the minority population of *IL33*-expressing hybrid MECs is present in the adult human breast, and the percentage of *IL33+* cells increases with age.

### IL33 induces basal differentiation and growth in HMECs

To functionally test whether IL33 impacts differentiation of primary human mammary epithelial cells (HMECs) as seen in mice, we treated cells with IL33 in 2D and 3D. IL33 treatment increased the organoid formation capacity of HMEC cells (Fig. 7d, e), suggesting that IL33 has similar functions in primary mouse and human mammary epithelial cells. Flow cytometry analysis of both 2D and 3D cultures treated with IL33 showed an increase in the percentage of CD49f + /EpCAM- (basal-like) cells and a decrease in the percentage of CD49f-/EpCAM+ (luminal-like) cells (Fig. 7f–h, Supplementary Fig. 13a). Furthermore, we find that HMECs treated with IL33 in both 2D and 3D culture had an increased percentage of CD44+ cells, a marker for immature stem-like cells in breast cancer (Fig. 7i–k, Supplementary Fig. 13b). Thus, our analysis reveals a minority population of *IL33+* cells in the human mammary gland that expresses luminal and basal markers, and IL33 treatment induces expression of basal markers in HMECs.

## Discussion

Our study lays the groundwork for understanding the complex relationship between aging and pregnancy in the mammary gland, such as basal-cell bias and differentiation. We report that pregnancy not only normalizes the age-induced expansion of basal cells but also concurrently reduces the capacity of basal and luminal cells to form organoids. Luminal cells in aged parous mice retain an involution-specific signature as seen in young parous mice, which could potentially make them more susceptible to immune surveillance[27,55–57].

Hybrid MECs expressing luminal and basal genes have been identified previously and are associated with a less-differentiated phenotype, higher plasticity, and are capable of tumor initiation[58–61], but precise molecular signatures of these cells are not fully defined. Our work identifies a previously unknown *IL33+* hybrid MEC population that accumulates with age in the basal layer of the mouse mammary gland. The majority of studies on breast tumor initiation implicate luminal cells as the cell of origin[62], that lose lineage integrity to express basal markers during tumor initiation and with aging[13–16]. Our analysis does not conclusively determine the cell of origin for the hybrid cells (basal →luminal or luminal→basal). However, given the location of the hybrid cells, we speculate that basal cells may acquire hybrid features with aging by turning on pathways active in luminal cells during early pubertal development. In support of this model, studies have shown that basal cells can acquire hybrid phenotypes upon transformation[63] and form luminal-like tumors[58,60], which are the predominant subtypes

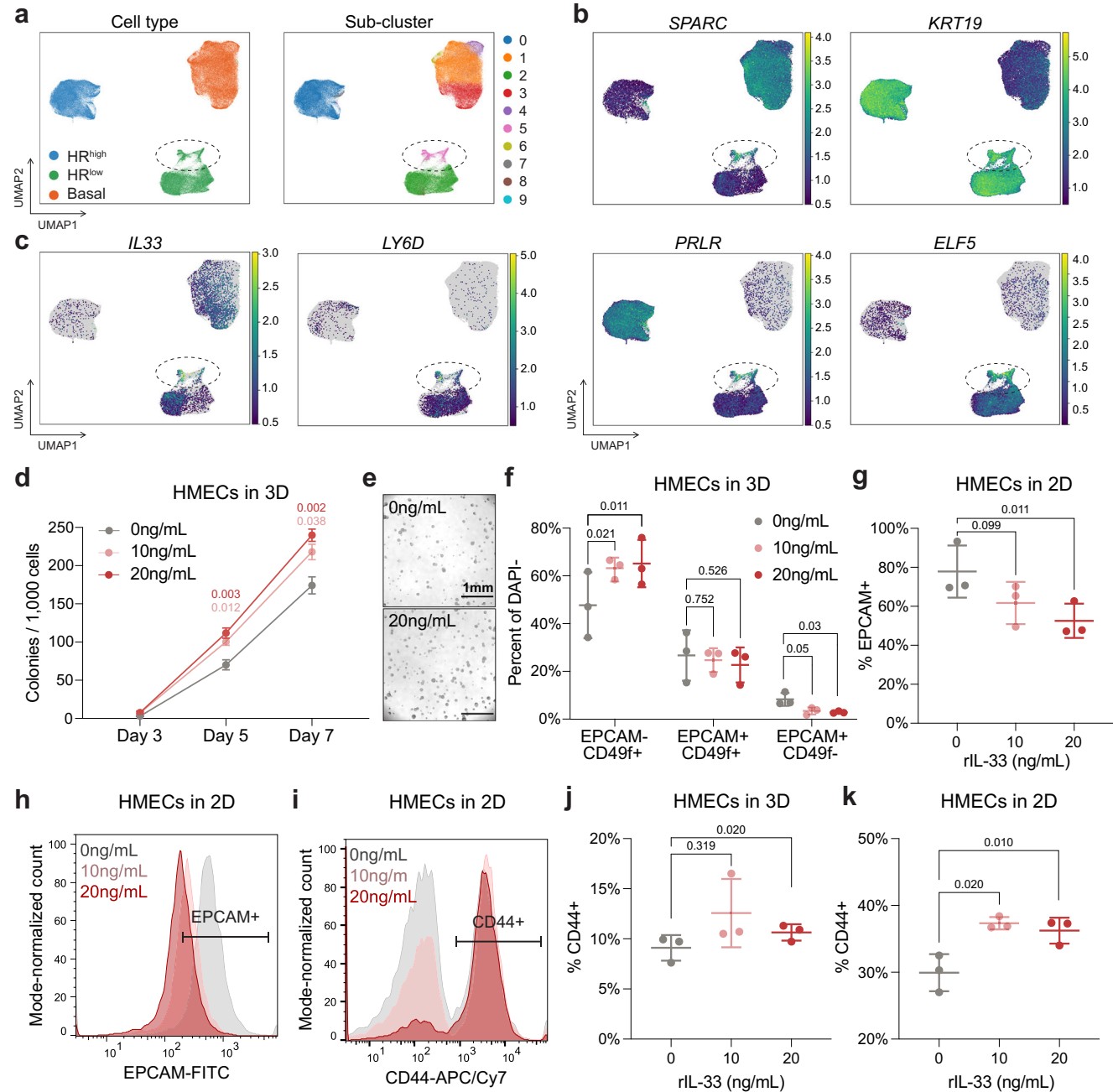

**Fig. 7 | *IL33* is expressed in a minority population of hybrid cells in the human breast and induces aberrant proliferation and differentiation in HMECs.**
**a** UMAP plots of human mammary epithelial cells (GSE235326) colored according to annotated cell type (left) and subcluster (right). **b** UMAP plots colored by expression of basal (*SPARC*) and luminal (*KRT19*, *PRLR*, and *ELF5*) gene markers. **c** UMAP plots of hybrid gene markers (*IL33* and *LY6D*). Cells with no expression of the gene are plotted in gray for visualization purposes. **d** Number of HMEC colonies over 7 days in culture supplemented with 0, 10, or 20 ng/mL IL33 (*n* = 3 independent experiments, 2 technical replicates per experiment). **e** Representative images of HMECs grown in 3D and exposed to 0, 10, or 20 ng/mL IL33 after 7 days in culture. **f** Percentage of EPCAM-CD49f + , EPCAM+CD49f + , EPCAM+CD49f- cells in 3D HMEC cultures exposed to 0, 10, or 20 ng/mL IL33 after 7 days in culture (*n* = 3 independent experiments). **g** Percentage of EPCAM+ cells in 2D HMEC cultures exposed to 0, 10, or 20 ng/mL IL33 after 7 days in culture (*n* = 3 independent experiments). **h, i** Representative histograms of flow cytometry data depicting EpCAM+ (**h**) and CD44+ (**i**) cells from IL33-treated HMECs. **j, k** Percentage of CD44+ cells in HMECs grown in 3D (**j**) and 2D (**k**) exposed to 0, 10, or 20 ng/mL IL33 after 7 days in culture (*n* = 3 independent experiments). Statistical significance was determined by performing two-way ANOVA with Geisser-Greenhouse correction and Tukey's multiple comparison test (**d**, **f**) or paired two-sided Student's t-tests (**g**, **j**, **k**). Scale bar reflects 1 mm. Data are presented as mean values +/- S.E.M (**d**) or +/-S.D. (**f**, **g**, **j**, **k**). Source data are provided as a Source data file.

found in postmenopausal breast cancers[64]. Future work will determine the precise contribution of *Il33*+ hybrid cells in the basal layer to tumor initiation with aging.

The postnatal mammary gland development and maintenance are largely driven by lineage-committed progenitor cells that expand during pregnancy to form the milk-producing cells[65–67]. However, rare bipotent mammary stem cells in the basal layer can expand with pregnancy and contribute to the alveolar lineage[67,68]. We speculate that during pregnancy, some basal cells either turn on an alveolar gene expression program or directly contribute to the alveolar lineage, leading to a decrease in basal cells and the hybrid MEC population in aged parous mice. Supporting this hypothesis, our results show an

increase in the quiescent hormone-responsive stem cell marker TSPAN8[26] with aging that is reduced in aged parous mice.

Our work identifies a minority population of *IL33*-expressing hybrid MECs present in the adult human breast, as well. However, our analysis in the human breast with respect to pregnancy remains inconclusive. The majority of human breast scRNA-seq datasets/atlases (such as the HBCA[51] and integrated Human Breast Cell Atlas, iHBCA[22]) do not include the age at first pregnancy and/or time since last birth, which may impact the accumulation of the hybrid population. Moreover, as the majority of the aged samples available in published datasets have previously undergone a pregnancy, it significantly reduces the number of nulliparous samples (particularly those > 50 years of age) that are available to analyze. In the singular dataset that does include age at first pregnancy (Reed et al.[22]), there is only one patient sample that does not carry mutations in breast cancer susceptibility genes, therefore preventing us from rigorously testing any hypotheses. However, our functional data using human primary epithelial cells demonstrates that IL33 can induce proliferation and basal-cell differentiation in vitro.

IL33 has been extensively studied in inflammation and immune cell responses[69], but its role in epithelial cell plasticity is unknown. Our results show that IL33 promotes basal organoid formation, stimulates epithelial proliferation, and induces aberrant ductal morphology in vivo. While our in vitro data confirms the ability of IL33 to directly act on basal epithelial organoids, we cannot rule out the potential cross-talk with the microenvironment, such as immune cells or fibroblasts, that can impact MECs. We also find that IL33 induces KRT6A expression in basal cells in vitro, as seen in the aged nulliparous mammary gland, while in young mice, IL33 induces expression of KRT6A in luminal cells in vivo. We speculate that this difference is because the basal cells in vivo are largely non-proliferative and do not give rise to luminal cells under homeostatic conditions. However, in the organoid culture, the basal cells are proliferative and differentiate to give rise to luminal cells. Therefore, when the transition from basal to luminal in vitro occurs in the presence of IL33, it induces expression of *Krt6a*. Notably, when basal cells acquire hybrid signatures due to expression of activated *Pik3ca*[H1047R][60] or ablation of the luminal layer[70], they dramatically increase the expression of *Il33*[60,70]. Transplantation of basal cells with inducible constitutively active PIK3CA results in glandular adenosis characterized by dilated ducts[47], similar to our IL33-treated mice. Moreover, mammary glands with ectopic overexpression of FGF3, which feature identical ductal dilation phenotypes, were found to display defective growth regulation via skewed mitogenic and apoptotic signals[48]. Importantly, the FGF3-driven ductal dilation and hyperplasia are reversible and were resolved by withdrawal of FGF3, similar to our 8dpt group, in which the ductal dilation had effectively recovered. We hypothesize that the natural aging process in nulliparous mammary glands involves a gradual, sustained exposure to IL33 in comparison to our short 5-day exposure in vivo model. Thus, in our model, the effects of IL33 are resolved 8 days post-treatment. While our study can partially recapitulate some aspects of this phenotype in young mice with IL33 treatment, the age-related accumulation and aberrant differentiation cannot be fully recapitulated with short-term IL33 treatment in vivo. Further studies will be necessary to fully resolve the role of IL33 in MECs with aging and in tumorigenesis.

Recent work has uncovered a role for IL33 in pancreatic cancer, where *Il33* expression is induced post-injury and cooperates with mutant *Kras* to promote neoplastic transformation[46,49]. Moreover, studies have demonstrated a cell-intrinsic role of IL33 in B-cell development[71], repair during skin[72] and lung[73] injury, and promoting inflammation in chronic pancreatitis[74]. Notably, in the context of lung injury, as basal epithelial stem cells activate expression of *Il33* to initiate proliferation, the *Krt6a* promoter activity increases—an

observation that was not recapitulated in *Il33*-/- mice[73]. These studies, combined with our data, suggest that *Il33* expression promotes a cell-state that is proliferative and plastic, allowing for injury repair or reprogramming during transformation. Future studies will determine the role of *Il33* during mammary gland development, in establishing a hybrid cell state, and tumorigenesis.

# Methods

## Mice
All mice used for this study were maintained at the UCSC Animal Facility/Vivarium in accordance with the guidelines of the Institutional Animal Care and Use Committee (Protocol #Sikas2311dn).

Wild-type C57BL/6 mice (young), retired breeders, and age-matched mice when appropriate were purchased from Charles River Laboratories and The Jackson Laboratory. Simultaneously, an aging colony was maintained in-house. Only female mice were used in the study, as male mice do not undergo pregnancy. The age of mice is indicated in the figures and figure legends. All mice were housed in climate-controlled rooms with a minimum air exchange rate of eight times per hour. The ambient temperature was maintained between 20 and 24 °C, and the relative humidity ranged from 45 to 65%. An automated 12:12 h light-dark cycle was implemented in all animal holding rooms. The mice were recorded for estrous, and 18 M mice were found to be non-cycling. Mice are euthanized by $CO_2$ followed by cervical dislocation.

## H&E staining and quantification
H&E staining was performed as previously described[75]. Briefly, mammary glands were dissected from parous and nulliparous mice, fixed in 4% PFA, and paraffin-embedded for histology. Sections underwent a standard staining protocol, with 5-min incubation in 100% xylene and ethanol (100%, 70%) followed by DI water for deparaffinization. Sections were stained with Hematoxylin for 30 s, then bluing solution ($NH_3OH$ + MilliQ H2O, 2 min) and Eosin for 1 min, followed by dehydration and mounting (Permount). Visualization of mammary gland ducts was performed using the Leica Widefield microscope, and ductal quantification was conducted using Fiji.

## Mammary gland whole mounts and branching analysis
Inguinal mammary glands were removed, adhered to a Superfrost Plus Microscope slide (Fisherbrand Cat. No. 1255015), and incubated in Carnoy's Fixative overnight at room temperature. Mammary glands were then incubated in 70% ethanol, 50% ethanol, and then DI water for 15 min each. Fixed mammary glands were stained overnight with Carmine Alum at room temperature, then dehydrated in 15-min incubations in increasing ethanol grades, followed by clearing in xylene and mounted in Permount.

Images of the entire mammary gland wholemount for both parous and nulliparous samples were imported as a TIFF file onto the Fiji/ImageJ software, where a three-by-three tiled ROI was chosen from a consistent distance from the lymph node. The ROI was processed to remove background noise. The image was skeletonized, and the branches dilated to improve clarity. A Sholl analysis (neuroanatomy) plugin was used on the skeletonized image, and a vertical radius from the side of the image was set as a consistent measurement throughout all samples to account for decreased branching normally seen at the edges of an image. After setting the starting radius as zero microns and selecting the best-fitting Sholl methods, the Sholl analysis program was run. This generated a "Sholl Log Plot," which indicated branching intersections over an area, as well as a "Sholl Regression Coefficient (k)", which described branching complexity. A k value closer to zero indicated a higher branching complexity. A "Sholl Mask (heat map)" was also generated to visually depict regions of higher branching complexity, where the red areas indicated maximal branching.

## Immunofluorescence staining of mammary gland paraffin sections

Mammary glands were fixed in 4% paraformaldehyde (PFA) in PBS and embedded in paraffin for immunostaining. 5 μm sections were deparaffinized, dehydrated, and autoclaved for 15 min in Tris-EDTA buffer [10 mM Tris 1 mM EDTA (pH 9.0)] for antigen retrieval. Tissue sections were incubated overnight at 4 °C with primary antibodies diluted in tris-buffered saline (TBS) + 5% bovine serum albumin (BSA) (antibodies are listed in Supplementary Table 2). Samples were subsequently washed (2X) with TBS + 0.05% Tween for 10 min and were incubated with donkey anti-rat Alexa Flour 647 (1:400), donkey anti-chicken Alexa Flour 488 (1:400), donkey anti-rabbit Alexa Flour 594 (1:400) conjugated secondary antibodies (Jackson ImmunoResearch Laboratories) in TBS + 5% BSA 0.1% Tween for 1 h at room temperature (RT). Samples were subsequently washed (3×) with TBS + 0.1% Tween and were incubated with DAPI in TBS + 5% BSA + 0.1% Tween (1:10,000) for 10 min. All the immunofluorescence sections and cells were mounted in Fluoromount-G (Genesee). Images were acquired by a Solamere Spinning Disk Confocal microscope. Images were processed using Fiji.

## Immunofluorescence staining of primary MEC organoids

Organoids were collected at day 12–14 in culture and recovered from the growth factor-reduced Matrigel by incubating with Cell Recovery Solution (Corning) for 1 h on ice with gentle shaking. 5 mL collection tubes and pipette tips were precoated in PBS + 5% BSA. Organoids were washed with cold PBS (2×) before being fixed with 4% PFA in PBS for 45 min on ice. Fixed organoids were then washed in PBS (1×) and incubated with 0.2% glycine in PBS for 20 min at room temperature. After a final wash, fixed organoids were resuspended in 0.05% sodium azide in PBS and stored at 4 °C.

Organoids were permeabilized in cold 100% methanol for 10 min on ice, washed in PBS (1×), and incubated in blocking buffer (0.1% BSA, 0.3% Triton-X, 5% normal donkey serum) for 3 h at room temperature with light shaking, followed by overnight incubation with primary antibodies in blocking buffer at 4 °C with light shaking. The next day, they were washed (3×) in PBS + 0.3% Triton-X and incubated overnight with secondary antibodies in a blocking buffer at 4 °C with light shaking in the dark. Organoids were again washed (3×) in PBS + 0.3% Triton-X and incubated with DAPI for 10 min at room temperature. After a final round of washes (3X), organoids were transferred to μ-Slide 8 Well Glass Bottom slide (Ibidi) pre-coated in poly-L-lysine (Sigma-Aldrich) and imaged on a Spinning Disk Confocal microscope. Images were processed using Fiji.

## Organoid assays

Primary MECs were sorted into complete organoid media as previously described[75,76] (Advanced DMEM F/12, 10% FBS, 1% PSA, 50 ng/mL EGF, 100 ng/mL Noggin, 250 ng/mL R-Spondin-1, 1X N2, 1X B27, 1X Gluta-MAX, 10 mM HEPES) supplemented with Y-27632 (10uM). 1000 cells were plated in 96-well ultra-low attachment plates seeded with a 50 μL mixture of growth factor-reduced Matrigel and irradiated L-Wnt3a-secreting-3T3 feeder cells (11,000 feeder cells per 50uL growth factor-reduced Matrigel). Organoids were cultured at 37 °C, 5% CO2 with added humidity. On day 4, Y-27632 supplemented media was removed and replaced with complete organoid media. Fresh media was added every other day thereafter. Basal and luminal organoids were imaged at day 6 and 10 in culture, respectively, on a Zeiss Live Cell microscope and analyzed on Biodoc.AI to collect data on organoid count and average size. For IL33 treatment experiments, organoids were cultured as described above with the addition of IL33 (0 ng/mL, 5 ng/mL, 10 ng/mL).

## Mammary gland digestion and processing

L2-5 and R2-5 mammary glands were harvested, minced, and chemically digested overnight in Advanced DMEM F/12 with 1% PSA, gentle collagenase/hyaluronidase, and DNAse I at 37 °C, 5% CO2, and added humidity as previously described[77]. Briefly, partially digested glands were then mechanically digested by pipetting with a serological pipette until no tissue pieces were visible. Digested glands were washed with staining buffer (Hank's Balanced Salt Solution, 2% Bovine Calf Serum, 1% PSA) and centrifuged (300 × g) at 4 °C for 5 min. Red blood cells were lysed with 5 mL of ACK Lysis buffer for 5 min, and cells were washed with 15 mL of staining buffer. Cells were treated with 0.25% Trypsin with EDTA and gently pipetted continuously for 2–3 min to digest the basement membrane. Cells were then treated with DNAse I and Dispase and pipetted continuously for 2–3 min to prevent clumping. The single-cell suspension was then filtered through a 40 μm mesh strainer and pelleted via centrifugation (300 × g, 4 °C, 5 min). Cells were then resuspended in staining buffer and transferred to FACS tubes for staining.

## Fluorescence-activated cell sorting (FACS)

All antibodies used in the study have been previously validated by the manufacturer or by previous studies. Cells were stained with antibodies listed in Supplementary Table 2 for 15 min at room temperature, as previously described[75,76]. Stained cells were then washed with a staining buffer, resuspended with DAPI (1:10,000), filtered, and analyzed on a BD Biosciences FACSAria cell sorter. See Supplementary Fig. 2 for FACS gating strategies. Data were analyzed using FlowJo software (10.10.0).

## Lentiviral Transduction

For *Trp53* knockdown experiments, pSicoR-GFP-sh*Trp53* was a gift from Tyler Jacks (Addgene plasmid # 12090; http://n2t.net/addgene: 12090; RRID:Addgene_12090[45]). Viruses were produced in HEK293T cells using the second- generation lentiviral system and transfection using Lipofectamine 2000 (Life Technologies) as previously described[77]. Supernatants were collected at 48 h, filtered with a 0.45 μm filter, and precipitated with lentivirus precipitation solution (Alstem LLC) per the manufacturer's instructions. Viral titers were determined by flow cytometry analyses of HEK293T cells infected with serial dilutions of concentrated virus.

## IL-33 treatment of 3-month nulliparous mammary glands

Three-month-old nulliparous C57BL/6 mice were purchased from The Jackson Laboratory. Mice were injected peritoneally daily with 1 μg of mouse recombinant IL-33 (Catalog number 580504, R&D Systems) or PBS for five consecutive days as previously described[46]. Mice were humanely euthanized in accordance with the guidelines of the Institutional Animal Care and Use Committee (Protocol #Sikas2311dn), and mammary glands were collected 1 and 8 days after treatment for analysis.

## EdU labeling and detection by flow cytometry

Wild-type 12-week-old C57BL/6 female mice were purchased from The Jackson Laboratory (#000664). An 8 mg/mL stock EdU solution was prepared by dissolving EdU (Life Technologies, #A10044) in sterile PBS (Life Technologies, #14190250). 10 mg/kg EdU was delivered via intraperitoneal (I.P.) injection into each mouse daily, as illustrated in Fig. 6a. After processing the mammary gland tissue to a single cell suspension, EdU was detected according to the manufacturer's instructions (Life Technologies, Click-iT Plus EdU Alexa Fluor 488 Flow Cytometry Assay Kit #C10632). Since EdU detection requires fixed cells, samples were stained with ZombieUV (Biolegend #423107) according to the manufacturer's instructions prior to fixation in order to assess viability. Samples were filtered and analyzed on a BD Biosciences FACSAria cell sorter.

## Annexin-V cell death flow cytometry analysis

Following antibody labeling and washes, cell pellets were resuspended in Annexin-V Binding Buffer (Biolegend #422201) to reach a

concentration of $1 \times 10^6$ cells/mL. 5 μL of AnnexinV-PE (Biolegend, #640908) was added for every 100 uL of single cell suspension. Samples were incubated at room temperature away from light for 15 min before washing with Annexin-V Binding Buffer. Phases of cell death were determined by DAPI and AnnexinV signal (see Supplementary Fig. 10).

## Primary human mammary epithelial cell (HMEC) culture
HMECs were purchased from ATCC (PCS-600-010), who authenticates and validates the cells, and cultured according to manufacturer instructions. All experiments involving HMECs were performed using low-passage cells (<P6). For 3D HMEC culture, 1000 HMECs were seeded in a 96-well ultra-low attachment plate on top of a layer of Corning™ Matrigel™ GFR Basement Membrane Matrix (Catalog number CB40230C). Colonies were grown in Mammary Epithelial Cell Basal Medium (ATCC PCS-600-030) supplemented with Mammary Epithelial Cell Growth Kit (ATCC PCS-600-040) and 2% GFR Matrigel. 2D and 3D cell cultures were maintained for seven days before analysis on a BD Biosciences FACSAria cell sorter.

## Single-cell RNA-seq cell preparation and sequencing
Libraries were generated using the 10× Genomics Chromium Next GEM Single Cell 5' Reagent Kits v2 (Dual Index) (cat#1000264) following the manufacturer's protocol. Single-cell suspensions were prepared from mammary glands as described above and encapsulated using the Chromium Controller to generate Gel Beads-in-Emulsion (GEMs), allowing for barcoding of individual cells. Post-GEM generation, reverse transcription (GEM-RT) was performed within each GEM, followed by cleanup and cDNA amplification. The cDNA underwent fragmentation, end repair, and A-tailing. Dual index adapters were ligated to the cDNA fragments, which were then amplified using PCR. The libraries were quantified using a Qubit Flex Fluorometer (Invitrogen Q33327), and their size distribution was assessed using an Agilent TapeStation 4200. Sequencing libraries were constructed to target 200 million paired-end reads (400 million total) to achieve a coverage of 20,000 reads per cell from a targeted capture of 10,000 cells. Sequencing was performed on an Illumina NovaSeq 6000 following the manufacturer's recommendations.

## Single-cell data preprocessing and quality control
Sequencing data were processed using the 10× Genomics Cell Ranger pipeline, with reads aligned to the mouse reference genome (mm10). Quality control steps included removing: (1) genes detected in fewer than 3 cells, (2) cells with <600 or >8000 genes, (3) cells with total UMI counts <2000 or > 12,000, and (4) cells with >1.5% mitochondrial gene content. The mitochondrial gene ratio was calculated as the percentage of UMIs mapped to mitochondrial genes relative to total cellular UMIs. Doublets were identified and removed using Scrublet[78], excluding 723 cells. All preprocessing and downstream analyses were performed using Scanpy (v1.11.0).

## Single-cell data integration, dimensionality reduction, and annotation
We integrated our single-cell dataset with a pre-processed 18-month nulliparous dataset from Tabula Muris Senis[29]. UMI counts were normalized to a target sum of 1e4 per cell, log-transformed, with a pseudocount of 1. The combined dataset included 18,341 cells and 26,056 genes. Principal component analysis (PCA) was performed for dimensionality reduction, and batch effects were corrected using Harmony (v0.0.4)[79]. Neighborhood graphs were computed on the corrected data and used for Leiden[80] clustering (resolution = 0.80), identifying sixteen distinct clusters. Clusters were manually annotated based on established marker genes. Data visualization was performed using Uniform Manifold Approximation and Projection (sc.tl.umap)[81].

## Cell cycle analysis and epithelial gene scoring
Cell cycle phase classification was performed using the Regev lab's curated cell cycle gene set, assigning each cell to its respective phase[82]. To classify epithelial cells, we used established marker genes for basal, HR-high luminal, and HR-low luminal cells based on transcriptomic profiles of adult MECs[30]. Enrichment scores for these markers were computed across four epithelial populations: basal, HR-high luminal, HR-low luminal, and hybrid MECs. Fibroblasts and T cells (gray) are included as controls. Both cell cycle and epithelial subtype scoring were carried out using the sc.tl.score_genes function from the Scanpy library[83].

## Pseudotime inference and cellular potency prediction
Pseudotime analysis was conducted on epithelial subtypes (basal, HR-low, HR-high, and hybrid). Partition-based graph abstraction (PAGA, v1.2) was performed using Scanpy (v1.9.6), with diffusion maps computed using basal cells as the root. To validate trajectory inference, pseudotime was also estimated using pyslingshot (v0.1.3), the Python implementation of Slingshot, again using basal cells as the starting node. Slingshot pseudotime values were scaled between 0 and 1 for comparison with PAGA results. Cellular potency was assessed using CytoTRACE 2 (v1.0.0) in Python. Visualizations were generated with Scanpy, Seaborn, and Matplotlib. To evaluate differences in pseudotime between epithelial subtypes, we performed pairwise Wilcoxon Rank Sum tests, applying Benjamini-Hochberg correction to control the false discovery rate. Statistical tests were implemented using scipy.stats.mannwhitneyu (SciPy v1.11.1) and multipletests from statsmodels.stats.multitest.

## Analysis of published single-cell RNA sequencing data
We investigated the presence of hybrid cells in previously published and publicly available single-cell datasets as outlined in Supplementary Table 1. We analyzed three mouse and two human datasets. We used previously published processed single-cell data where available. Else, single-cell RNA sequencing data were processed and visualized using Scanpy (version 1.11.0) in Python. Following preprocessing and normalization, principal component analysis was performed for dimensionality reduction, followed by the construction of a nearest neighbor graph. Uniform Manifold Approximation and Projection (UMAP) embeddings were generated using the sc.tl.umap function to visualize cellular populations in two dimensions. We performed Leiden clustering at resolutions [0.02, 0.05, 0.1, 0.5, 2.0] and annotated and filtered for epithelial cells (luminal and basal) for downstream analysis. Hybrid proportions were computed for each Leiden cluster as the (number of cells expressing *Il33* and *Krt6a*/total cells in Leiden cluster) and plotted using seaborn's boxplot function. In the human dataset, cells with no expression of the gene are plotted on the UMAP in gray.

## Statistics and reproducibility statement
All graphs display the average as central values, and error bars indicate ± SD unless otherwise indicated. P values are calculated using paired or unpaired t-test, ANOVA, Wilcoxon rank-sum test, and Mann–Whitney U test, as indicated in the figure legends. All P and Q values were calculated using Prism (10.2.2) or Python (3.11.5), unless otherwise stated. For animal studies, sample size was not predetermined to ensure adequate power to detect a prespecified effect size; no animals were excluded from analyses, experiments were not randomized, and investigators were not blinded to group allocation during experiments.

## Reporting summary
Further information on research design is available in the Nature Portfolio Reporting Summary linked to this article.

## Data availability

Data generated or analyzed during this study are included in this published article (and its supplemental information files). Data needed to evaluate the conclusions in the paper are present in the paper and/or the Supplemental Materials. Single-cell RNA-sequencing data generated in this study has been deposited in the Gene Expression Omnibus (GEO) with the primary accession code GSE272932 and can be explored at https://aging-mouse-pregnancy.cells.ucsc.edu. The published single-cell RNA-sequencing data used in this study are available in the GEO database under accession codes GSE216542, GSE205573, GSM2967054, and GSE195665, and in the ArrayExpress database under the accession code E-MTAB-13664. Source data are provided with this paper.

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

## Acknowledgements

We thank Bari Nazario and Patricia Lovelace for their help with flow cytometry. The FACS Aria instrument was funded by NIH grant 1S10RR02933801 and Cytoflex RRID SCR_021149 and grant NIH S10OD030423. We thank Benjamin Abrams, UCSC Life Sciences Microscopy Center, RRID: SCR_021135 for technical support during image acquisition and processing. We thank Gunsagar Singh Gulati for advice on data integration and Bryce Manso for flow cytometry analysis of immune cells. We also thank the animal facility core members for animal maintenance. We thank Camilla Forsberg, Lindsay Hinck, Aaron Newman, and members of the Sikandar lab for critical feedback on the manuscript. The authors declare no competing interests. This work was supported by the Hellman Fellows Award and startup funds (to S.S.S), and NIH T-32 (5T32GM133391-04) and NIH/NCI F31 (1F31CA294932-01A1) (to A.O). S.S.S is also supported by the NIH/NCI (R37CA269754).

## Author contributions

S.S.S. and A.O. conceived and designed the study. A.O. performed most of the experiments and analyzed the data with assistance from V.H.A, M.D., and S.K., and under the supervision of S.S.S. V.H.A. collected samples for single-cell RNA sequencing and performed experiments with human cells. P.M. performed all bioinformatic analysis under supervision of S.S.S A.O., and S.S.S. wrote the manuscript with contributions from P.M. and V.H.A. All authors commented on the manuscript.

## Competing interests

The authors declare no competing interests.
