## [Transparent Peer Review file · Nature Communications]

Divergent aging of nulliparous and parous mammary glands reveals IL33+ hybrid epithelial cells.

Corresponding Author: Dr Shaheen Sikandar

Version 0:

Reviewer comments:

Reviewer #1

(Remarks to the Author)

In this study Olander and colleagues have investigated cellular and molecular changes induced by pregnancy in mice. They analyzed mammary glands of 3-, 8-, and 18-month-old virgin or multiparous mice by flow cytometry to assess the relative proportion of luminal/basal cells and also performed RNA-seq on FACS sorted basal and luminal cells. Subsequently, they also performed single cell RNA-seq on mammary glands from 18-month-old virgin and multiparous mice. Lastly, they tested the effect of IL33, a gene they identified as differentially expressed in mammary epithelial cells 18-month-old virgin and multiparous mice. Overall, the manuscript has limited data and novelty.

Specific points:

1. Full term pregnancy in early adulthood only reduces the risk of postmenopausal ER+ breast cancer, while the risk of triple-negative breast cancer (TNBC) is actually increased. Mice do not get ER+ mammary tumors, thus, they may not be the best model to use to assess the effect of pregnancy on ER+ breast cancer risk.
2. There have been prior related studies in mice that compared retired breeders (old parous) and old virgin mice and also virgin mice and different stages of pregnancy/lactation. For example: <https://pubmed.ncbi.nlm.nih.gov/25619437/> and scRNA-seq profiling (cited by the authors): <https://pubmed.ncbi.nlm.nih.gov/34187545/>
3. Similarly, the increase in hybrid basal-luminal cells with age has been described by multiple groups, both in human and mice, and IL33 has been shown to promote cancer stem cells in breast cancer.
4. The authors have to include 3-month-old parous mice in their flow cytometry and RNAseq analyses (Figure 1-3) and also 3-month-old virgin and parous to see how quickly the differences due to pregnancy are observed. This is important, because the authors argue that their study is unique, since they looked at old parous mice compared to prior studies that analyzed mice shortly after pregnancy. But, w/o showing data for 3-month-old virgin and parous this is not possible to conclude.
5. Similarly, in Figure 4 the authors argue that IL33 treatment of organoids from young mice phenocopies aging. But they do not analyze organoids from old mice and also do not compare 3-month-old virgin vs. parous. These additional samples would be critical to include.

Reviewer #2

(Remarks to the Author)

This study by Olander et al. investigates the combined effects of aging and pregnancy on mouse mammary epithelial cells (MECs). The authors report that pregnancy reduces the age-induced expansion of basal cells and partially normalizes the basal:luminal cell ratio in aged mice. They also report the presence of a small population of IL33-expressing hybrid MECs,

which accumulate in aged nulliparous mice, but reduced in aged parous mice. The authors show that pregnancy induces transcriptomic changes in both basal and luminal cells, promoting a differentiated state and reversing some age-associated transcriptional programs. Finally, the authors show that IL33 treatment of young basal cells mimics the aged nulliparous MECs, increasing the proportion of basal cells and inducing a KRT6a+ hybrid state, which may confer a survival advantage for cells with oncogenic mutations. Overall, the study suggests that pregnancy-induced changes in MECs contribute to long-term breast cancer protection by maintaining lineage integrity and reducing the accumulation of potentially tumorigenic hybrid cells. The study is well presented and written. Generally, there is a slight over interpretation of some of the data which should be substantiated with further evidence. Below are the main points this reviewer believes should be addressed.

- 1) Title: I personally feel that the title might be slightly overreaching, as the paper does not explicitly demonstrate that pregnancy reduces the formation of the hybrid progenitor. The findings just show that the hybrid progenitor is identified in the old nulliparous samples but is less prevalent in the old parous samples.
- 2) Figure 2 claims that pregnancy "also reverses aged-associated transcriptional programs". The authors have done DGE analysis to directly search for genes specific to that group (as far as I understand - they should specify the exact comparisons they do to generate these DGEs. i.e. are the genes for 18m NP a result of the test: 18m NP vs (3m NP + 18m P)? I suspect this is the case and thus forces the result that they claim. For the claim that Parity is undoing the effects of ageing, I would instead want to see how the genes for a 3m NP vs 18m NP test compares to those of the 18m NP vs 18m P. And potentially how they might oppose each other.
- 3) Fig 2f, Again I am not sure this GO analysis has really shown this? To me it is just descriptive of the DEGs with each state but does not show evidence of parity undoing ages effects?
- 4) Fig 3, It would be interesting to see the distribution of hybrid cells across samples. Are they primarily identified in specific samples, or are they generally upregulated in all aged NP samples?
- 5) Fig 3. On the tabula muris data, it would be good to include graphs showing the integration, along with more detailed information on the distribution of cells across the samples. With this if they are claiming to have found a completely new cell type I would want them to find it in other datasets or provide some explanation why no one else has seen these cells?
- 6) Fig3, I really want to see doublet scores for each of the clusters/cells calculated (scDbfFinder, scrublet, or any other comparable tool). Additionally, why are there no pericytes/vascular mural cells in their dataset?
- 7) Fig3d, the authors should consider adding a control plot here for fibroblasts (for example) to see what this analysis would look like for an unrelated population. Additionally I would want to know if these hybrid cells have any unique gene markers. This would help convince me they are a distinct cell type not some doublet/conglomeration of the two.
- 8) Fig 4, the results are interesting BUT I'd really like to see what this treatment would do in the mice - does it do the same in vivo with a fully functioning immune system present?
- 9) Finally, the authors need to perform a cross species analysis utilising the large human breast cell atlas datasets now available to see if these findings in the mouse do translate to the human.

Reviewer #3

(Remarks to the Author)

In this manuscript, Dr. Shaheen Sikandar and colleagues use RNAseq and scRNA techniques as well as organoid and immunostaining to investigate the cell and molecular changes in aged parous mammary glands compared to aged nulliparous glands as well as young nulliparous glands. This is an interesting developmental study considering both pregnancy and ageing that has not been well studied previously. The significant findings include the identification of a hybrid cell population that expands with ageing in nulliparous mice but this expansion is prevented in the parous aged group. In addition, the finding of elevated IL33 expression in this hybrid population is also interesting since IL33 has been implicated in cancer risk and the authors showed convincing data that IL33 treatment can promote organoid growth and hybrid cell expansion, mimicking ageing effects. However, there are also a number of significant weaknesses including over-interpretation of IL33 experiments on basal vs. luminal cells, premature experiments in testing IL33 and cancer relevance, and lack of adequate controls for some of the experiments. In addition, while data from the IL33 treatment of organoids provide an exciting and mechanistic link between ageing and hybrid cells, an in vivo validation, which should not take too long, would provide a much stronger argument. Detailed comments are listed below:

1. Fig 1: An increase in the ratio of basal cells vs. luminal cells does not prove an increase of the basal population. It could be due to the drop of the luminal population, which seems to be the case here (Fig 1a). This ratio also neglects the change in the overall epithelial content in the comparing groups. Therefore, besides presenting the plots showing the luminal basal ratio and ER+ vs. ER- luminal ratio, please plot the basal and luminal population sizes for all three groups of mice. Please also make sure that the FACS images are representative of the plotted data.
2. Fig 1: Without these absolute population size comparisons, the authors cannot conclude "the age-induced expansion of basal cells is normalized by pregnancy, the luminal cells retain a residual involution program, with an increased proportion of CD14+ HR low luminal cells."

3. Fig 2: it is shocking and highly interesting that their gene expression data on FACS sorted basal cells suggest parity leads to production of milk associated genes. However, since FACS sorting cannot guarantee no luminal cell contamination, it is critical that the authors confirm this suggestion by performing spatial assays such as IF or IHC to RNAscope to demonstrate the milk gene expression in the basal layer in aged glands.

4. Paragraph starting line 123: please reference 18m B in the statement "We found significant enrichment of genes involved in cytoskeletal remodeling, extracellular matrix, and contractile functions (Fig. 2e)." The authors conclude this paragraph with the statement that "These data suggest that pregnancy normalizes age-induced transcriptional changes in luminal and basal epithelial cells, while also inducing a differentiated state." But the data presented in this paragraph does not logically lead to this conclusion.

5. IL33 treatment of 3m NP basal cells in organoids led to blocked luminal differentiation based on flow cytometry (Fig 4b), but no change of the proportion of K8+ luminal cells (Fig S6F). These two data seem to be contradictory.

6. While IL33 treatment did not affect 3m NP luminal cells in forming organoids, this data does not mean that IL33 treatment would not affect other features such as cell differentiation and expression of K6 or K8 or K5. Please analyze these luminal derived organoids in the same way as the basal cell derived organoids were characterized. If the luminal organoids do not expand the K6+ population, the data would strength the claim of basal origin of the hybrid cell population.

7. Studying IL33 impact on cells with Tp53 knockdown helps increase the cancer implication of this developmental work. However, this experiment is highly preliminary. Only one shRNA was used, and it lacks the confirmation of Tp53 knockdown. Also, non-targeting shRNA controls are missing but would provide the critical baseline comparison to understand the specific impact of p53 loss. In the absence of these data, it is premature and misleading to suggest that IL33 has a p53-dependent impact.

8. There are no adequate and decisive data to conclude that the K6+ hybrid cells originate in the basal population as the authors claimed in Discussion.

9. While the IL33 treatment of organoids experiments provide an exciting and mechanistic link between ageing and hybrid cells, an in vivo validation, which should not take too long, would provide a much stronger argument.

Version 1:

Reviewer comments:

Reviewer #1

(Remarks to the Author)

The authors have responded to each of the points raised by the reviewers and included additional experiments to strengthen the study. The revised manuscript is improved, but still there are some remaining concerns.

The single cell RNA-seq analyses are still suboptimal, there are much more sophisticated data analysis and presentation tools than UMAP and bar plots.

Many bar plots lack statistical tests and p-values (like Fig 3 g,h), also it's better to indicate actual p-values instead of asterisks for significance.

The in vivo experiment with IL33 (new Figure 6) is useful, but it raises several new issues:

(1) IL33 acts on many other cell types, not just epithelial cells, so are the observed changes through IL33 directly acting on epithelial cells or changing the microenvironment?

(2) The authors only treated 3 mice and only for 5 days – this is too few mice to ensure reproducibility and significant and they need an explanation for how only 5 days of treatment has such dramatic effects on mammary ductal morphology. Was this due to proliferation or apoptosis? The authors should test this.

(3) similarly, in the scRNAseq data (Fig 7 1-b) it would be important to plot the Ki67+ cells, since the authors claim that IL33 induces aberrant proliferation.

Reviewer #2

(Remarks to the Author)

I thank the authors for addressing the questions raised. The manuscript is much improved.

Two minor comments:

1) The authors should use the integrated HBCA (Reed et al 2024 Nat. Genetics) to assess the expression of IL33. Its a much larger dataset which integrated the 7 main human scRNAseq studies (2.1 million cells). They should also show if these cells are represented more or less in any of the datasets.

2) The discussion starts with "Our study resolves and comprehensively maps the complex relationship between aging and pregnancy in the mammary gland, such as basal-cell bias and differentiation." I don't believe the current study fully resolves or comprehensively maps this relationship. I think this needs to be toned down to reflect the data presented.

Reviewer #3

(Remarks to the Author)

The authors have been responsive to the critiques and this revision has improved many aspects of the previous data

presented. Figs 1-5 are excellent and present a great coherent story.

In response to reviewers asking for in vivo confirmation of IL33 effect, and for human validation, they added figures 6 and 7. However, these two new figures caused more concerns than solving problems.

IL-33-induced changes in mammary glands in mice at day 1 post completion of 5 days of treatment are not consistent with the cell culture and organoid data presented before this new figure 6 - KRT6A expression was induced in luminal cells but not basal cells but the reserve was observed in the in vitro experiments. More convincing explanations should be provided for this inconsistency. Furthermore, the authors did not detect an impact of IL-33 treatment at day 8, calling into question about the critical role of IL-33 in mediating the ageing effect on expanding the hybrid cell population. The authors should explain why the lack of detecting a more long-term change did not weaken their central conclusions.

Human scRNA data did not detect a hybrid cell population that is consistent with their mouse studies, and they did not detect any impact of ageing or parity on this cell population, calling into the question of whether their analysis was appropriate or proper datasets were used. In the absence of more convincing data, this figure probably should be dropped.

Version 2:

Reviewer comments:

Reviewer #1

(Remarks to the Author)

The authors have responded to each of the remaining comments of the reviewers by revising the manuscript, including adding some additional experiments.

However, the dramatic increase in duct size after just 1 day of IL33 administration is not resolved conclusively (Fig 1b, f). The authors checked proliferation and apoptosis and found that IL33 mainly increases proliferation. However, to see such a dramatic (at least 2-fold) increase in duct size in 24 hrs would mean that virtually all cells in the duct double in 24 hrs – which is not matching their EdU labeling data showing that 10-20% of the basal and luminal cells have incorporated EdU in ~6 days. There is clearly an increase in proliferation, but the magnitude and time course of this does not match the dramatic phenotypic change that occurs in a day. Have the authors considered alternative explanations like swelling of the ducts – checked cell size and potential fluid accumulation?

Reviewer #2

(Remarks to the Author)

Thank you for addressing my concerns. The analysis of the iHBCA presented in the rebuttal is informative and should be part of the main paper.

I have no further comments.

Reviewer #3

(Remarks to the Author)

The authors have been responsible to the reviewer's comments. The revised manuscript now include additional data and revised presentation to address the inconsistency between in vitro finding and in vivo validation and bioinformatic studies of published human scRNAseq data. Overall, the manuscript has strengthened, but the inconsistency moderates the significance of the study. Additional concerns are detailed below:

1. Fig 2b: the proportion plot shows the 18M P group has a lower proportion of all cell types. If the same total cell numbers were analyzed in 18M P vs 18M NP, reduction of some cell groups should be accompanied by decrease of others. In this study, nearly twice as many cells were used in the P group compared to NP group (NP, red, 3,327 cells vs P, blue, 7,305 cells), to this reviewer, it is mathematically impossible that the 18M P has lower counts for all cell subsets. As a result, all the statements regarding this fig are a bit shaky. To help readers, the authors should ensure that the same number of cells were used for this comparison in Fig 2A and 2B. Also since 3 samples are used, intra-sample variations should be shown.
2. Fig 2a/b: The integration with Tabula Muris (red, n = 2,993 cells) is confusing. If more cells were indeed needed, the authors should sequence more cells from their own experiments. Integrated analysis should be secondary not primary, and readers should be convinced that the public data by others was comparable.
3. Fig 4a/b: this figure is used to show KRT6A+ cells are switched from the luminal layer to the basal cell layer with ageing. While the 18M NP panel shows convincing basal location of KRT6a+ cells, the 3M NP panel does not show convincing luminal cell staining for KRT6a.
4. Fig 5H: the organoid culture of Tp53 WT control basal cells does not show response to IL33. This seems to be inconsistent with Fig 5A.
5. Line 262-264: The authors state that "Wholemount analysis of the mammary glands from IL33-treated mice showed a significantly increased lymph node area and dramatic dilation of the ducts at 1dpt (Fig. 6b, c)." But the phrase "increase lymph node area" is confusing. Do the authors mean increased ducts around the lymph node area? If so, an explanation is needed for this difference around the LN area compared to other parts of the mammary glands.
6. Line 266-268: "Aside from LN area, analysis of mice at 8dpt showed no significant changes in any of the parameters above, suggesting the effects of short term IL33 treatment were transient (Fig. 6b-f)." This sentence does not make sense. Do the author mean changes are seen around the LN area, but not other parts of glands? If so, explanations are again needed.
7. Fig 6I and J: IL33 treatment for 1 day (but now 8 days) led to a change of the ratio of basal vs. luminal cells. But this

potentially important observation is not stated in the results section nor discussed.

Version 3:

Reviewer comments:

Reviewer #1

(Remarks to the Author)

The authors response to the remaining comment is fine. Could they also add the statement that expansion of the ducts due to swelling cannot be excluded to the manuscript also when they are describing the data? the enlargement of the ducts is a very dramatic phenotype (even if it was 48 hrs not 24 hrs treatment) so I'm sure others will also question this.

The responses to reviewer 3 and revision seem OK to me, although I do not see the figures to make sure resolution is publishable quality- but this you can check during QC.

Point by point response.

Original comments are in blue, authors replies are in black.
Changes are highlighted in green in the revised manuscript.

Please note: Figures included in the revised manuscript paper are referenced in the point-by-point response but not included in this document. Only figures specific to the point-by-point response are included here.

Reviewer #1 (Remarks to the Author):

In this study Olander and colleagues have investigated cellular and molecular changes induced by pregnancy in mice. They analyzed mammary glands of 3-, 8-, and 18-month-old virgin or multiparous mice by flow cytometry to assess the relative proportion of luminal/basal cells and also performed RNA-seq on FACS sorted basal and luminal cells. Subsequently, they also performed single cell RNA-seq on mammary glands from 18-month-old virgin and multiparous mice. Lastly, they tested the effect of IL33, a gene they identified as differentially expressed in mammary epithelial cells 18-month-old virgin and multiparous mice. Overall, the manuscript has limited data and novelty.

We thank the reviewer for their comments and assessment of our manuscript. We have significantly expanded the scope of our work, adding several new figures in the main and extended data as per the suggestions from all the reviewers. We believe that our revised work provides an analysis of aging and pregnancy that has not been done before.

Specific points:

1. Full term pregnancy in early adulthood only reduces the risk of postmenopausal ER+ breast cancer, while the risk of triple-negative breast cancer (TNBC) is actually increased. Mice do not get ER+ mammary tumors, thus, they may not be the best model to use to assess the effect of pregnancy on ER+ breast cancer risk.

We appreciate the reviewer's insightful comment and welcome the opportunity to clarify our rationale. The goal of our study was to investigate how pregnancy influences the aging of stem/progenitor cells in the mammary gland, specifically examining whether certain subsets are enriched or lost with aging and pregnancy. Previous studies have long utilized mice as a model system to study parity-induced protection^{1,2} and understand underlying mechanisms³⁻⁵. More recently studies have used mice to study aging in mammary epithelial cells and their relation to breast cancer risk⁶⁻⁸. Thus, we followed past precedent in using mice as a model system to understand the long-term impacts of pregnancy.

As the reviewer correctly noted, the risk of breast cancer increases in the first 10 years following pregnancy in humans. The key limitation of previous studies is that they looked at the post-pregnancy mammary gland when there is increased risk and it is not post-menopausal (perimenopause in mice)³⁻⁵. This makes our study unique, as it specifically examines a time point beyond this 10-year window⁹⁻¹², aiming to better mimic conditions relevant to human biology—where women who experience an early first pregnancy gain protective benefits much later in life, typically after the age of 50. Moreover, none of the aging studies in mammary epithelial cells account for pregnancy⁶⁻⁸.

While it is true that mice usually get basal like breast cancer there are mouse models of ER+ breast cancer that have been used to study human disease. Some of these include *ESR1*, *CCDN1*, prolactin, and *WNT1* overexpression, *PIK3CA* gain of function and loss of *STAT1* –

comprehensive reviews are here^{13,14}. As one recent example published in *Nature Cancer* in 2025, a group from Genentech generated and utilized a mouse model to study human ESR1 mutations arising due to resistance to ER α -targeted therapies in breast cancer^{15,16}. They showed that the cells present in the mammary gland of mice carry the ESR1 mutation that are resistant to endocrine suppression have a distinct gene expression pattern that is between luminal and basal cells. We have analyzed the single cell RNA-sequencing data from this manuscript¹⁵ (**Fig. 3i**). We find that the cells arising from ESR1 mutant mice after endocrine suppression (labelled as endocrine suppressed mutant enriched or ESME cells in the published manuscript) are enriched in expression of *Krt6a/Il33* (**Fig. 3j,k**).

Multiple studies have also employed mouse ER+ breast cancer cell lines, which were previously characterized¹⁷ and recently used to understand metastatic dissemination of ER+ breast cancer¹⁸. Lastly, a recent study showed that the same mutation that generates ER- (basal-like) tumors in young mice gives rise to ER+ tumors in aged mice¹⁹. Since many of the earlier studies introducing mutations to generate breast cancer have been done with young mice, it is possible that if these experiments were repeated with aged mice, they would give rise to ER+ breast cancer.

Finally, as requested by Reviewer #2, we have extended our findings to human datasets and primary mammary epithelial cells. These results are now included as a new figure (**Fig. 7**) in the main manuscript. Briefly, our findings identify a minority population of cells with a similar gene expression pattern as seen in nulliparous aged mice (**Fig. 7a-c, Extended Data Fig. 10**). We also treated primary human mammary epithelial cells (HMECs) with IL33 and observed results consistent with basal epithelial cells from the mouse mammary gland. Specifically, IL33 enhances organoid formation capacity, restricts luminal differentiation, and promotes the expression of the cancer stem cell marker CD44²⁰ (**Fig. 7d-k**). In summary, we believe that our model system is highly relevant to studying the effects of aging and pregnancy on stem/progenitor cells in the mammary gland.

2. There have been prior related studies in mice that compared retired breeders (old parous) and old virgin mice and also virgin mice and different stages of pregnancy/lactation. For example: <https://pubmed.ncbi.nlm.nih.gov/25619437/> and scRNA-seq profiling (cited by the authors): <https://pubmed.ncbi.nlm.nih.gov/34187545/>

We thank the reviewer for the comment. As mentioned above, these studies were done at an early age (28-36 weeks, 9 months in mice, ~30-38 human years), which is not post-menopausal. Moreover, the first study cited by the reviewer was done using bulk-RNA sequencing obscuring cellular heterogeneity²¹ and second one was done immediately post-involution where there is an increase in breast cancer risk²². Therefore, while these studies have informed our analysis, they do not address our main objective: to characterize the molecular and cellular effects of pregnancy on the aged mammary gland. A more detailed description of both studies is given below.

*Huh et al.*²¹ interrogated three experimental groups: virgin (3, 9, 24wk), pregnant (D10, 16, 19), and retired breeder (28-36wk). Their analysis incorporated bulk DNA methylation and RNA sequencing, obscuring cellular heterogeneity and identification of minority cell populations. Moreover, they discovered “*that pregnancy had the most significant effects on CD24(+)CD61(+)CD29(hi) and CD24(+)CD61(+)CD29(lo) cells, inducing distinct epigenetic states that were maintained through life*”. This conclusion is incomplete, as their oldest point sampled is 36wk/9m, which translates to ~30-38 human years. Given that most breast cancer patients receive a diagnosis over a decade later, we feel that our time point of 18m (~56 years in humans) is a more comprehensive approach to address our objective.

The second study *Pal B et al.*²² explore the developing mammary gland using scRNA-seq of 8 key stages spanning E18.5 to 10wk virgin and 3wk post-involution. This narrow time frame precludes any interrogations into the effects of aging on the mammary gland, which is a key component and source of novelty for our study. In our study we have used this dataset to show that *Il33+* and *Krt6a+* cells are present in the terminal end bud during development (**Fig. 3h**), but the proportion of these cells decreases in the adult and post involution (**Fig. 3h**).

As requested by the reviewer in point #4, we also integrated the data from *Pal B, et al.*²² with our data to directly compare all the groups. We find that while there is a decrease in the proportion of *Il33+/Krt6a+* cells right after pregnancy, the population is very small in young mice to begin with, it increases with age and is reduced in aged parous mice (**Fig. 3g, Fig. 4a, b**).

3. Similarly, the increase in hybrid basal-luminal cells with age has been described by multiple groups, both in human and mice, and IL33 has been shown to promote cancer stem cells in breast cancer.

We thank the reviewer for their comment. While the effects of aging on loss of lineage integrity in luminal cells have been described²³, no study has assessed how early pregnancy affects the aging process or the mechanisms leading to loss of lineage integrity with age.

We for the first time show that age induced accumulation of hybrid cells located in the basal layer and demonstrate that a previous pregnancy reduces this accumulation. We also show that IL33 mimics the age-induced loss of lineage integrity in young basal cells. Moreover, in a previous study²³ *Vatter et al.* propose that luminal cells acquire basal features whereas our study suggests that basal cells could acquire luminal features. This is an important distinction because while it has been thought that luminal cells are the cell of origin for breast cancers recent studies have showed that basal cells can be the cell of origin^{24,25}.

Regarding IL33, its role in normal mammary stem/progenitor cells, cellular plasticity and tumor initiation have never been studied. Previous studies have primarily examined the role of IL33 from the microenvironment on tamoxifen resistance and metastasis^{26,27}. Therefore, our finding that **1.** in a minority population of cells, IL33 expression increases with age **2.** is reduced in aged parous mice and **3.** regulates growth and differentiation of normal mammary epithelial cells to our knowledge have not been described before.

4. The authors have to include 3-month-old parous mice in their flow cytometry and RNAseq analyses (Figure 1-3) and also 3-month-old virgin and parous to see how quickly the differences due to pregnancy are observed. This is important, because the authors argue that their study is unique, since they looked at old parous mice compared to prior studies that analyzed mice shortly after pregnancy. But, w/o showing data for 3-month-old virgin and parous this is not possible to conclude.

We thank the reviewer for this comment. As per the reviewer's suggestion, we have now revised Fig. 1 & 3 to include a 3-month-old parous mice (mice underwent a minimum of two pregnancies and were analyzed at least 3 weeks post-involution).

In brief, we repeated all the experiments in Fig. 1 with a 3-month-old parous (3M P) mouse side by side with 3M NP, 18M NP and 18M NP. Our experiments characterize the mammary gland at these different stages through flowcytometry and organoid assays, which has never been done before. We find that parity has distinct effects on the aging of luminal and basal cells – **1.** Luminal cells from parous mice do not undergo age related loss that is seen in nulliparous mice **2.** Parity restricts basal cell expansion that is seen with aging, but it is not a complete rescue. We also performed organoid assays with the 4 groups to find that while aging increase organoid formation capacity of basal cells, it reduces the organoid forming capacity of luminal cells.

For Fig 3., we were able to compute the fraction of *Krt6a+//33+* cells across the 4 different groups as described above and included in the revised manuscript (**Fig. 3g**). We find that while there is a decrease in the *Krt6a+//33+* cells immediately post-pregnancy, this percentage is significantly lower in 3M NP and 3M P mice (**Fig 3g**). Thus, our data suggests that accumulation of the *Krt6a+//33+* cells occurs with aging in nulliparous mice. Moreover, we have confirmed this analysis using immunofluorescence (**Fig. 4a-c**).

For the previous Fig. 2, we integrated the *Pal et al.* dataset²² with ours and performed pseudo bulk analysis across all the groups (young nulliparous, young parous, aged nulliparous and aged parous). We have included this analysis below as part of the point-by-point response (**Fig. R1**).

First, we confirmed successful isolation of basal and luminal cell populations from the pseudobulk analysis by expression of *Krt14* and *Acta2* in basal samples and *Krt8* and *Krt18* in luminal samples (**Fig. R1a**). To identify transcriptomic differences between nulliparous and parous MECs as they age, we performed differential gene expression analyses on aged basal or luminal cells versus young basal or luminal cells from either parous or nulliparous mice (i.e., 18M NP vs. 3M NP and 18M P vs. 3M P) and identified differentially expressed genes with opposing log2 fold changes (**Fig. R1b**). Basal cells from parous mice displayed increased expression of the basal cell lineage marker *Lipg*²⁸ and ECM-related genes (*Ltbp2*, *Col4a1*, **Fig. R1c, d**) with age, whereas basal cells from nulliparous mice showed a decrease in expression with age. In agreement with previous studies²⁹, basal cells from nulliparous mice had an age-dependent increase in expression of *Arg1*. However, we find that basal cells from parous mice showed an opposite trend, with *Arg1* being highly expressed in basal cells from young parous mice and returning to levels similar to young nulliparous basal cells in aged parous basal cells (**Fig. R1c, d**). Similarly, the expression of *Tspan8*, which has been previously reported to mark a population of quiescent hormone responsive stem-cells³⁰, increased with age in nulliparous basal cells but this was reversed in parous (**Fig. R1c, d**). We had independently analyzed and confirmed the age- and parity-dependent trend in *Tspan8* expression at the protein level with flow cytometry analysis (**Extended Data Fig. 3**).

The number of age-specific differentially expressed genes (DEGs) with opposing log2 fold changes between parous and nulliparous mice was far greater in luminal cells compared to basal cells (**Fig. R1e**), underscoring the dramatic impact of pregnancy on luminal cell transcriptomes. As was observed in basal cells, *Arg1* expression increased with age in nulliparous mice, but decreased with age in parous mice (**Fig. R1e, f**). Other genes showing similar expression patterns with age and parity include *Kitl*, *Sparc*, *Serpina3n*, and *Igfbp3*, which are often associated with the basal cell lineage²⁸, as well as the basal progenitor cell marker *Cd36*³¹ (**Fig. R1e, f**).

Our findings support the hypothesis that MECs display a loss of lineage integrity over the course of aging and suggest that previous pregnancies attenuate this age-dependent phenotype while instilling a persistent transcriptomic memory of pregnancy. However, we feel this analysis is largely descriptive as pointed out by Reviewer 2 and tangential to the main point of the paper. We have therefore removed it from the main manuscript.

[FIGURE REDACTED]

Figure R1: Parous and Nulliparous MECs display distinct transcriptomic patterns with aging. (a) Expression of basal (*Krt14*, *Acta2*) and luminal (*Krt18*, *Krt8*) lineage markers in bulk RNA-seq samples collected from FACS-sorted basal and luminal cells of 18M NP (red) and 18M P (blue) mammary glands. 18M bulk RNA-seq samples were integrated with 3M NP (grey) and 3M P (green) scRNA-seq datasets (GSE103275) for pseudo-bulk RNA-seq analyses. (b) Schematic diagram describing the strategy used to identify differentially expressed genes with

aging (18M NP vs. 3M NP and 18M P vs. 3M P) and exhibit opposing log₂ fold changes (Log₂FC). (c) Differentially expressed genes (DEGs) in 18M NP basal vs. 3M NP basal (pink) and 18M P basal vs. 3M P basal (blue) with opposing Log₂FC values. DEGs were filtered for $p_{\text{adj.}} < 0.05$ and $|\text{Log}_2\text{FC}(\text{NP}) - \text{Log}_2\text{FC}(\text{P})| > 1.5$. (d) Expression values of select genes identified in (c) across comparison groups. (e) Differentially expressed genes (DEGs) in 18M NP luminal vs. 3M NP luminal (pink) and 18M P luminal vs. 3M P luminal (blue) with opposing Log₂FC values. DEGs were filtered for $p_{\text{adj.}} < 0.05$ and $|\text{Log}_2\text{FC}(\text{NP}) - \text{Log}_2\text{FC}(\text{P})| > 1.5$. (f) Expression values of select genes identified in (d) across comparison groups. Gene expression is represented as transcript per million (TPM) $\times 10^3$ (a) or normalized values from DESeq2 (d, f). $n = 3$ mice per group

5. Similarly, in Figure 4 the authors argue that IL33 treatment of organoids from young mice phenocopies aging. But they do not analyze organoids from old mice and also do not compare 3-month-old virgin vs. parous. These additional samples would be critical to include.

We thank the reviewer for their comment. As suggested by the reviewer we have included all the 4 groups using both the mammary glands from mice (Fig. 4a-c) and on organoids derived from them (Fig. 4d-g).

Consistent with the single cell RNA-sequencing data, we find that the 18M NP mice have the highest percentage of KRT6A⁺ cells and that this percentage is reduced in 18M P mice. Moreover, we find that there is a slight but significant reduction of KRT6A⁺ cells in the young parous mice (Fig. 4a, b). For the *in vitro* culture, we observe the increase in KRT6A⁺ cells in the organoids derived from basal cells (Fig. 4d, e) but do not see any differences in organoids derived from luminal cells (Fig. 4f, g) in 18M NP mice.

Reviewer #2 (Remarks to the Author):

This study by Olander et al. investigates the combined effects of aging and pregnancy on mouse mammary epithelial cells (MECs). The authors report that pregnancy reduces the age-induced expansion of basal cells and partially normalizes the basal:luminal cell ratio in aged mice. They also report the presence of a small population of Il33-expressing hybrid MECs, which accumulate in aged nulliparous mice, but reduced in aged parous mice. The authors show that pregnancy induces transcriptomic changes in both basal and luminal cells, promoting a differentiated state and reversing some age-associated transcriptional programs. Finally, the authors show that IL33 treatment of young basal cells mimics the aged nulliparous MECs, increasing the proportion of basal cells and inducing a KRT6a⁺ hybrid state, which may confer a survival advantage for cells with oncogenic mutations. Overall, the study suggests that pregnancy-induced changes in MECs contribute to long-term breast cancer protection by maintaining lineage integrity and reducing the accumulation of potentially tumorigenic hybrid cells. The study is well presented and written. Generally, there is a slight over interpretation of some of the data which should be substantiated with further evidence. Below are the main points this reviewer believes should be addressed.

We thank the reviewer for their assessment of our manuscript and constructive feedback. We have addressed the concerns of the reviewer below by including new data.

1) Title: I personally feel that the title might be slightly overreaching, as the paper does not explicitly demonstrate that pregnancy reduces the formation of the hybrid progenitor. The findings just show that the hybrid progenitor is identified in the old nulliparous samples but is less prevalent in the old parous samples.

We appreciate the reviewer for raising this concern. In view of the reviewer's concern, we have revised our title to read, "*Characterization of the mammary gland with pregnancy and aging*

identifies a population of IL33+ hybrid progenitors." We believe that this title more accurately reflects the scope of our work and hope that the reviewer agrees with us.

Our initial rationale for the title was that the only distinction between the 18-month parous and nulliparous groups is whether one group experienced pregnancy at an early age. Therefore, we do believe that pregnancy/parity is the key variable influencing the formation of hybrid progenitors. Moreover, our analysis of *Pal B, et al.* dataset²², shows that while the percentage of *Krt6a+//IL33+* cells decreases immediately after pregnancy, it is significantly lower in both 3-month-old nulliparous (3M NP) and parous (3M P) mice compared to 18-month-old nulliparous (18M NP) mice (**Fig. 3g, Fig. 4a,b**). We find that the 18-month parous (18M P) mice exhibit substantially fewer *Krt6a+//IL33+* cells as compared to 18-month nulliparous mice (18M NP) (**Fig. 3g, Fig. 4a,b**). These findings suggest that the accumulation of *Krt6a+//IL33+* cells occurs with aging but is mitigated in mice that have undergone pregnancy.

2) Figure 2 claims that pregnancy "also reverses aged-associated transcriptional programs". The authors have done DGE analysis to directly search for genes specific to that group (as far as I understand - they should specify the exact comparisons they do to generate these DGEs. I.e. are the genes for 18m NP a result of the test: 18m NP vs (3m NP + 18m P)? I suspect this is the case and thus forces the result that they claim. For the claim that Parity is undoing the effects of ageing, I would instead want to see how the genes for a 3m NP vs 18m NP test compares to those of the 18m NP vs 18m P. And potentially how they might oppose each other.

We apologize for not being clearer about our methods and the figures. We had done exactly what the reviewer suggested in both cases. In response to reviewer #1, we have revised the figure to include 3-month old parous mice (**Fig. R1**). However, we believe that the analysis was tangential to the main point of the paper and have therefore removed it from the revised manuscript.

3) Fig 2f, Again I am not sure this GO analysis has really shown this? To me it is just descriptive of the DEGs with each state but does not show evidence of parity undoing ages effects?

We agree that the GO analysis was just descriptive. We have now removed the analysis to streamline the message in the manuscript. The analysis is included in the point-by-point response (**Fig. R1**).

4) Fig 3, It would be interesting to see the distribution of hybrid cells across samples. Are they primarily identified in specific samples, or are they generally upregulated in all aged NP samples?

We appreciate the reviewer for raising this concern. To address this, we have compared the distribution of hybrid cells in our dataset with the 10x Tabula Muris Senis dataset (**Extended Data Fig. 4d**). Interestingly, we find that this population is significantly higher in the 10x Tabula Muris Senis dataset compared to ours – Hybrid cells in 10x Tabula muris – 78 cells, 18M NP – 13 cells, 18M P– 3 cells (**Extended Data Fig. 4d**). In our initial clustering, we found 27 cells in the 18M NP mice in our dataset but with reclustering to identify pericytes as requested in Point #6, some of these cells clustered with the basal cluster (**Fig. R2, Extended Data Fig. 6b**). Importantly, we have now validated our findings using two independent aging datasets (**Fig. 3a-f**). We find that the number of cells in hybrid cluster is different across different experiments, but the population can be consistently identified in multiple datasets (see Point #5 below).

Fig. R2: UMAP plot of scRNA-seq data generated from the mammary glands annotated by cell-types (left). Cells previously annotated as hybrid are displayed in red (right).

Of note, our single-cell RNA sequencing was performed on three biological replicates using hashing to distinguish individual samples. However, due to technical issues, the hashing procedure did not function as expected, preventing us from deconvoluting the replicates.

Importantly, we have confirmed our analysis at the protein level through immunofluorescence staining for KRT6A across all 4 groups of mice (**Fig. 4a, b**) using at least 3-5 mice per group. Consistent with our single-cell RNA sequencing data, our immunofluorescence results show an accumulation of KRT6A+ cells in aged nulliparous mice (**Fig. 4a, b**). As requested by Reviewer 1 we also performed the staining on organoids derived from either basal or luminal cells in the 4 groups. We find that the basal organoids from the aged nulliparous mice have an increased proportion of KRT6A+ cells (**Fig. 4d, e**) but we see very few KRT6A+ cells in organoids derived from luminal cells (**Fig. 4f, g**). These protein-level findings further substantiate our single-cell RNA sequencing analysis.

5) Fig 3. On the tabula muris data, it would be good to include graphs showing the integration, along with more detailed information on the distribution of cells across the samples. With this if they are claiming to have found a completely new cell type I would want them to find it in other datasets or provide some explanation why no one else has seen these cells?

Thank you for bringing up this important point. Indeed, we can identify the *Krt6a+//l33+* population in the two other published 18M and 21M aging datasets (**Fig. 3a-f**), GSE216542⁶ (39 cells that we annotate as hybrid) and the SmartSeq data from Tabula Muris Senis (103 cells that we annotate as hybrid, *note – different dataset than 10x data*). Consistent with our analysis on the RNA and protein level, we find that *Krt6a+//l33+* cells are significantly higher in the aged nulliparous mice as compared to the young nulliparous mice (**Fig. 3a-f**).

We believe previous studies failed to detect this population because these cells were clustered with basal cells, as observed in **Fig. 3a, d**, classified as myoepithelial_8 and basal cell_12, respectively. The Tabula Muris Senis data are available on CellXGene and the GSE216542⁶ can be explored using the Shiny app provided in the paper.

Notably, 21-month and 18-month nulliparous mice exhibit the highest proportion of *Krt6a+//l33+* expressing cells within the basal-annotated subcluster (**Fig. 3c, f**). We think the key

reason we were able to detect this difference is that we compared nulliparous and parous glands at 18 months, revealing a significantly lower number of *Krt6a+//l33+* cells in the aged parous gland. Interestingly, as mentioned above in response to Reviewer 1, we find that the cells arising from ESR1 mutant mice after endocrine suppression (labelled as endocrine suppressed mutant enriched or ESME cells in the published manuscript¹⁵) are highly enriched in *Krt6a+//l33+* cells (**Fig. 3i-k**), suggesting that we have identified a disease relevant population that accumulates with aging.

6) Fig3, I really want to see doublet scores for each of the clusters/cells calculated (scDbfFinder, scrublet, or any other comparable tool). Additionally, why are there no pericytes/vascular mural cells in their dataset?

Doublet discrimination: We appreciate the reviewer's concern. All analyses presented in the original and current manuscript were performed after excluding doublets and any cells with abnormally high gene counts per cell, which are often indicative of doublets. We have included the UMAPs for doublet detection in the revised manuscript (**Extended Data Fig. 4a-c**).

However, it is important to note that doublet discrimination algorithms generate artificial doublets by simulating co-expression profiles from the gene expression matrix. Since these methods rely on a linear combination of gene expression profiles to predict doublets, it is possible that hybrid cells—defined by the expression of markers from both lineages—may receive a higher doublet score. Most importantly, we have now extended our validation to additional single-cell datasets (**Fig. 3**) and included protein-level validation (**Fig. 4**), confirming that this is a bona fide cell population that accumulates with aging and in ESR1 mutant mammary glands.

Pericytes and Vascular Mural Cells: We thank the reviewer for this comment. Initially we followed the Tabula Muris 10x data annotations for the different cell types to keep it consistent during integration. However, we have reclustered our data at a higher resolution and annotated pericytes as well as dendritic cells and included a revised UMAP and dot plot (**Fig. 2a-c, Extended Data Fig. 5a**). We find that pericytes cells are only a minority population residing between fibroblasts and endothelial cells as expected. We could not identify a specific cluster of vascular mural cells based on published markers. The recently published Angarola dataset (GSE216542⁶) with single cell RNA-sequencing for 3M and 18M mice also does not have an annotated cluster of vascular mural cells. We suspect that this is likely due to loss of cells during dissociation.

7) Fig3d, the authors should consider adding a control plot here for fibroblasts (for example) to see what this analysis would look like for an unrelated population. Additionally I would want to know if these hybrid cells have any unique gene markers. This would help convince me they are a distinct cell type not some doublet/conglomeration of the two.

Scoring for fibroblasts: As requested, we have added a control plot for fibroblasts and T-cells (**Fig. 2d**). This method has been previously used to determine lineage restriction in mammary development³². We find that fibroblasts score significantly higher with the basal gene signature because fibroblasts and basal cells both have mesenchymal features but score low with the luminal gene signatures (**Fig. 2d**). It is important to note that while some genes are exclusive to epithelial cells, certain markers are also expressed in non-epithelial cell populations. However, our analysis shows we can identify the populations using its specific gene signature.

Unique gene markers for the hybrid population: As the population is defined by co-expression of luminal and basal markers, there are limited genes exclusive to the hybrid cell population such as *l33*, *Aldh3a1*, *Dmkn*, *Dapl1*, *Ppp1r14c* but also expresses *Ly6d*, *Alcam*, *Dsc3*,

Lgals7, *S100a14*, *Anxa8* and *Atp1b1* (**Fig 2c**). We specifically chose to explore IL33 functionally, as it is exclusive to the hybrid population, expressed in the highest proportion of cells (**Fig. 2c**). Other studies that have identified hybrid populations through single cell RNA-sequencing often find that there is not a set of genes exclusive to the cluster but rather a gene expression signature^{15,33,34}.

8) Fig 4, the results are interesting BUT I'd really like to see what this treatment would do in the mice - does it do the same *in vivo* with a fully functioning immune system present?

We thank the reviewer for this suggestion. As suggested by the reviewer, we performed *in vivo* analysis as previously described in a model of pancreatic cancer³⁵ and included it in the revised manuscript (**Fig. 6**). Briefly, we find that short term IL33 treatment induces dramatic changes in ductal morphology (**Fig. 6b-g**), reduces the percentage of luminal cells as seen in aged mice (**Fig. 6h**) but does not increase the percentage of basal cells (**Fig. 6h**). We also profiled the major immune cell types in the mammary gland (**Extended Data Fig. 9a, b**). We find that there is an increase in the IL33 responder cells that are ST2 positive (**Extended Data Fig. 9c, d**) indicating that our treatment was specific to IL33, but no changes in other immune cell types (**Extended Data Fig. 9a, b**).

Importantly, we find that IL33 increases the percentage of KRT6A+ mammary epithelial cells *in vivo*, albeit exclusively in the luminal layer (**Fig. 6i, j**). This is an important distinction from our *in vitro* experiments where expression of IL33 does not affect luminal cells but impacts basal cell differentiation. We speculate that this distinction is because the basal cells *in vivo* are largely non-proliferative and do not give rise to luminal cells under homeostatic conditions. However, in the organoid culture the basal cells are proliferative and differentiate towards luminal cells. We speculate that when the transition from basal to luminal cells *in vitro* occurs in the presence of IL33, it induces expression of KRT6A.

Expansion of the basal layer that occurs with aging likely cannot be recapitulated with short-term IL33 treatment *in vivo* but will require the expression of IL33 using a basal specific promoter (KRT5 or KRT14) and we feel it is currently beyond the scope of this study. However, we believe that our manuscript identifies a population of IL33+ cells in the aged mammary gland, uncovering a previously unknown role for IL33 in MECs and opens a new area of investigation into IL33 in mammary epithelial biology.

9) Finally, the authors need to perform a cross species analysis utilising the large human breast cell atlas datasets now available to see if these findings in the mouse do translate to the human.

We thank the reviewer for this suggestion and have addressed this concern (**Fig. 7a-c and Extended Data Fig. 10**). We were able to identify a minority population of IL33+ cells that cluster with luminal HR^{low} cells but expresses markers of basal (*KRT5*, *SPARC*) and HR^{high} (*PRLR*) cells in the human breast cell atlas (GSE195665)³⁷ (**Fig. 7a, b** subcluster 5, pink, and **Extended Data Fig. 10**). Cluster 5 is also enriched in expression of other genes from our hybrid cluster such as *LY6D*, *DMKN*, *PPP1R14C* and *ALCAM* (**Fig. 7a-c and Extended Data Fig. 10b**). However, we were unable to identify robust expression of KRT6A (**Extended Data Fig. 10c**), albeit there is some, likely reflecting differences in mouse and human keratin markers.

We acknowledge that the expression of these genes is confined to a relatively small population of cells within the cluster. This may be due in part to technical limitations such as dropout events or low sequencing depth that specifically affects genes that are expressed at low levels. Furthermore, we did not observe differences in age or parity within this subcluster (**Extended Data Fig. 10d**). However, since the HBCA dataset does not include information on

age at first pregnancy or time since last birth, we are unable to assess whether this population declines in older women who experienced early pregnancy. Interestingly, cluster 5 shows higher representation of individuals with Asian and African American ancestry (**Extended Data Fig. 10d**), which may point to ancestry-associated biological differences.

Importantly, to understand whether IL33 has an effect in human mammary epithelial cells (HMECs), we performed organoid assays. We find that like organoids derived from basal cells in the mouse mammary gland, IL33 promotes organoid formation, increases the expression of CD49f+ and CD44 in HMECs (**Fig. 7d-k, Extended Data Fig. 11**).

Reviewer #3 (Remarks to the Author):

In this manuscript, Dr. Shaheen Sikandar and colleagues use RNAseq and scRNA techniques as well as organoid and immunostaining to investigate the cell and molecular changes in aged parous mammary glands compared to aged nulliparous glands as well as young nulliparous glands. This is an interesting developmental study considering both pregnancy and ageing that has not been well studied previously. The significant findings include the identification of a hybrid cell population that expands with ageing in nulliparous mice but this expansion is prevented in the parous aged group. In addition, the finding of elevated IL33 expression in this hybrid population is also interesting since IL33 has been implicated in cancer risk and the authors showed convincing data that IL33 treatment can promote organoid growth and hybrid cell expansion, mimicking ageing effects. However, there are also a number of significant weaknesses including over-interpretation of IL33 experiments on basal vs. luminal cells, premature experiments in testing IL33 and cancer relevance, and lack of adequate controls for some of the experiments. In addition, while data from the IL33 treatment of organoids provide an exciting and mechanistic link between ageing and hybrid cells, an *in vivo* validation, which should not take too long, would provide a much stronger argument. Detailed comments are listed below:

We warmly thank the Referee for nicely summarizing our study and for their appreciation that both pregnancy and aging have not been well studied previously.

1. Fig 1: An increase in the ratio of basal cells vs. luminal cells does not prove an increase of the basal population. It could be due to the drop of the luminal population, which seems to be the case here (Fig 1a). This ratio also neglects the change in the overall epithelial content in the comparing groups. Therefore, besides presenting the plots showing the luminal basal ratio and ER+ vs. ER- luminal ratio, please plot the basal and luminal population sizes for all three groups of mice. Please also make sure that the FACS images are representative of the plotted data.

2. Fig 1: Without these absolute population size comparisons, the authors cannot conclude “the age-induced expansion of basal cells is normalized by pregnancy, the luminal cells retain a residual involution program, with an increased proportion of CD14+ HR low luminal cells.”

We thank the reviewer for suggesting this method. We have addressed these comments by providing an analysis of the absolute values in the revised **Fig. 1** and in the new **Fig. 6**. The absolute values further highlight how pregnancy reduces some of the age-related shifts in the basal and luminal populations. We do not see a change in the overall epithelial compartment but continue to see an increase in the basal population (**Fig. 1c**). We also find that aged parous mice have a lower percentage of basal cells and a higher percentage of luminal cells (**Fig. 1c**).

Moreover, we see an increase in the HR-low luminal cells in young parous mice that is maintained with aging (**Fig. 1e**), confirming that luminal cells likely retain a residual involution program. As requested by Reviewer 1, we have expanded our findings to include data from 3M

parous mice (3 weeks post-involution) allowing us to compare the effects in young and old parous mice side by side.

3. Fig 2: it is shocking and highly interesting that their gene expression data on FACS sorted basal cells suggest parity leads to production of milk associated genes. However, since FACS sorting cannot guarantee no luminal cell contamination, it is critical that the authors confirm this suggestion by performing spatial assays such as IF or IHC to RNAscope to demonstrate the milk gene expression in the basal layer in aged glands.

We thank the reviewer for this observation. We have removed this analysis from the revised manuscript and included it only in the point-by-point response (**Fig. R1**). However, as suggested by the reviewer, we performed immunostaining for Csn1s1. We find that there is expression of Csn1s1 in rare aged parous basal cells (**Fig. R3**). Moreover, we find staining of Csn1s1 in aged nulliparous luminal cells (top row). However, we were unable to get any of the

available antibodies to work with immunofluorescence staining. Therefore, our analysis is based on the spatial localization of the cells rather than expression of basal keratin markers.

Figure R3: Representative images of immunohistochemistry stains against CSN1S1 in 18M nulliparous (top row) and 18M parous (bottom row) mammary gland sections. Arrow heads indicate CSN1S1+ cells localized to the basal layer. Scale bars reflect 250µm or 70µm.

4. Paragraph starting line 123: please reference 18m B in the statement “We found significant enrichment of genes involved in cytoskeletal remodeling, extracellular matrix, and contractile functions (Fig. 2e).” The authors conclude this paragraph with the statement that “These data suggest that pregnancy normalizes age-induced transcriptional changes in luminal and basal epithelial cells, while also inducing a differentiated state.” But the data presented in this paragraph does not logically lead to this conclusion.

We have now removed this analysis from the main manuscript but have included it in the point-by-point response to Reviewer #1 (**see Fig. R1**).

5. IL33 treatment of 3m NP basal cells in organoids led to blocked luminal differentiation based on flow cytometry (Fig 4b), but no change of the proportion of K8+ luminal cells (Fig S6F). These two data seem to be contradictory.

We thank the reviewer for raising this point – we do see a slight decrease in KRT8 staining in organoids treated with IL33 at the 20ng/ml dose, but it does not reach statistical significance

(**Fig. 5f, g**). We suspect our flowcytometry analysis is more robust and quantifiable than the 3D-imaging of the organoids. Moreover, some basal cells can also express KRT8 at low levels which may further confound the immunofluorescence staining in 3D.

6. While IL33 treatment did not affect 3m NP luminal cells in forming organoids, this data does not mean that IL33 treatment would not affect other features such as cell differentiation and expression of K6 or K8 or K5. Please analyze these luminal derived organoids in the same way as the basal cell derived organoids were characterized. If the luminal organoids do not expand the K6+ population, the data would strength the claim of basal origin of the hybrid cell population.

We completely agree with the reviewer, and we have run this analysis. We find that the luminal organoids treated with IL33 do not expand the KRT6A+ population (**Fig. 5g, bottom panel**) and have included the data in the revised manuscript.

7. Studying IL33 impact on cells with Tp53 knockdown helps increase the cancer implication of this developmental work. However, this experiment is highly preliminary. Only one shRNA was used, and it lacks the confirmation of Tp53 knockdown. Also, non-targeting shRNA controls are missing but would provide the critical baseline comparison to understand the specific impact of p53 loss. In the absence of these data, it is premature and misleading to suggest that IL33 has a p53-dependent impact.

We sincerely apologize for our oversight and the omission of the data from the original manuscript. The experiment was indeed run with a non-targeting shRNA control (**Fig. 5h**), and we had verified the knockdown of *Trp53* (**Extended Data Fig. 8**). We have revised the manuscript to include this data. We agree that our analysis is preliminary, but it demonstrates the impact of IL33 on proliferation of basal and luminal cells in the context of tumor promoting mutations. Our future work will expand on how IL33 promotes tumor initiation.

8. There are no adequate and decisive data to conclude that the K6+ hybrid cells originate in the basal population as the authors claimed in Discussion.

We thank the reviewer for bringing up this important point. Indeed, we speculated in the discussion that the KRT6A+ cells are likely to originate in the basal layer given our immunofluorescence data showing their localization in the basal layer. However, as per the reviewer's suggestion we have modified the discussion to state, "*Our analysis does not conclusively determine the cell of origin for the hybrid cells (basal →luminal or luminal→basal). However, as they are spatially located in the basal layer, we speculate that basal cells could acquire hybrid features with aging by turning on pathways active in luminal cells during early pubertal development*". We are happy to further refine the language if needed.

9. While the IL33 treatment of organoids experiments provide an exciting and mechanistic link between ageing and hybrid cells, an *in vivo* validation, which should not take too long, would provide a much stronger argument.

We thank the reviewer for this comment. As suggested by both Reviewers 2 & 3 (see point#8 for Reviewer 2), we performed the *in vivo* experiment and have included it in the revised manuscript (**Fig. 6**).

REFERENCES:

1. Medina, D. Mammary developmental fate and breast cancer risk. (2005) doi:10.1677/erc.1.00804.
2. Medina, D. & Smith, G. H. Chemical Carcinogen-Induced Tumorigenesis in Parous, Involuted Mouse Mammary Glands. *JNCI Journal of the National Cancer Institute* **91**, 967–969 (1999).
3. Feigman, M. J. *et al.* Pregnancy reprograms the epigenome of mammary epithelial cells and blocks the development of premalignant lesions. *Nat Commun* **11**, 2649 (2020).
4. Hanasoge Somasundara, A. V. *et al.* Parity-induced changes to mammary epithelial cells control NKT cell expansion and mammary oncogenesis. *Cell Rep* **37**, 110099 (2021).
5. Siwko, S. K. *et al.* Evidence that an early pregnancy causes a persistent decrease in the number of functional mammary epithelial stem cells—implications for pregnancy-induced protection against breast cancer. *Stem Cells* **26**, 3205–3209 (2008).
6. Angarola, B. L. *et al.* Comprehensive single-cell aging atlas of healthy mammary tissues reveals shared epigenomic and transcriptomic signatures of aging and cancer. *Nat Aging* **5**, 122–143 (2025).
7. Bai, H. *et al.* Progressive senescence programs induce intrinsic vulnerability to aging-related female breast cancer. *Nat Commun* **15**, 5154 (2024).
8. Li, C. M.-C. *et al.* Aging-Associated Alterations in Mammary Epithelia and Stroma Revealed by Single-Cell RNA Sequencing. *Cell Reports* **33**, (2020).
9. Goddard, E. T. *et al.* Association Between Postpartum Breast Cancer Diagnosis and Metastasis and the Clinical Features Underlying Risk. *JAMA Netw Open* **2**, e186997 (2019).
10. Borges, V. F., Lyons, T. R., Germain, D. & Schedin, P. Postpartum Involution and Cancer: An Opportunity for Targeted Breast Cancer Prevention and Treatments? *Cancer Research* **80**, 1790–1798 (2020).
11. Schedin, P. Pregnancy-associated breast cancer and metastasis. *Nat Rev Cancer* **6**, 281–291 (2006).
12. Lyons, T. R., Schedin, P. J. & Borges, V. F. Pregnancy and breast cancer: when they collide. *J Mammary Gland Biol Neoplasia* **14**, 87–98 (2009).
13. Borowsky, A. D. Choosing a Mouse Model: Experimental Biology in Context—The Utility and Limitations of Mouse Models of Breast Cancer. *Cold Spring Harb Perspect Biol* **3**, a009670 (2011).
14. Dabydeen, S. A. & Furth, P. A. Genetically engineered ER α -positive breast cancer mouse models. (2014) doi:10.1530/ERC-13-0512.
15. Liang, J. *et al.* ER α dysfunction caused by ESR1 mutations and therapeutic pressure promotes lineage plasticity in ER+ breast cancer. *Nat Cancer* 1–15 (2025) doi:10.1038/s43018-024-00898-8.
16. Liang, J. *et al.* Giredestrant reverses progesterone hypersensitivity driven by estrogen receptor mutations in breast cancer. *Sci. Transl. Med.* **14**, eabo5959 (2022).
17. Yang, Y. *et al.* Immunocompetent mouse allograft models for development of therapies to target breast cancer metastasis. *Oncotarget* **8**, 30621–30643 (2017).
18. Turrell, F. K. *et al.* Age-associated microenvironmental changes highlight the role of PDGF-C in ER+ breast cancer metastatic relapse. *Nat Cancer* **4**, 468–484 (2023).
19. Jenkins, E. C. *et al.* Age alters the oncogenic trajectory toward luminal mammary tumors that activate unfolded proteins responses. *Aging Cell* **21**, e13665 (2022).
20. Al-Hajj, M., Wicha, M. S., Benito-Hernandez, A., Morrison, S. J. & Clarke, M. F. Prospective identification of tumorigenic breast cancer cells. *Proceedings of the National Academy of Sciences* **100**, 3983–3988 (2003).
21. Huh, S. J. *et al.* Age- and Pregnancy-Associated DNA Methylation Changes in Mammary Epithelial Cells. *Stem Cell Reports* **4**, 297–311 (2015).
22. Pal, B. *et al.* Single cell transcriptome atlas of mouse mammary epithelial cells across development. *Breast Cancer Res* **23**, 69 (2021).
23. Vatter, F. A. P. *et al.* High-Dimensional Phenotyping Identifies Age-Emergent Cells in Human Mammary Epithelia. *Cell Reports* **23**, 1205–1219 (2018).

24. Koren, S. *et al.* PIK3CAH1047R induces multipotency and multi-lineage mammary tumours. *Nature* **525**, 114–118 (2015).
25. Van Keymeulen, A. *et al.* Reactivation of multipotency by oncogenic PIK3CA induces breast tumour heterogeneity. *Nature* **525**, 119–123 (2015).
26. Hu, H. *et al.* IL-33 facilitates endocrine resistance of breast cancer by inducing cancer stem cell properties. *Biochemical and Biophysical Research Communications* **485**, 643–650 (2017).
27. Shani, O. *et al.* Fibroblast-Derived IL33 Facilitates Breast Cancer Metastasis by Modifying the Immune Microenvironment and Driving Type 2 Immunity. *Cancer Research* **80**, 5317–5329 (2020).
28. Ortiz, J. R. *et al.* Single-Cell Transcription Mapping of Murine and Human Mammary Organoids Responses to Female Hormones. *J Mammary Gland Biol Neoplasia* **29**, 3 (2024).
29. Li, C. M.-C. *et al.* Aging-Associated Alterations in Mammary Epithelia and Stroma Revealed by Single-Cell RNA Sequencing. *Cell Reports* **33**, 108566 (2020).
30. Fu, N. Y. *et al.* Identification of quiescent and spatially restricted mammary stem cells that are hormone responsive. *Nat Cell Biol* **19**, 164–176 (2017).
31. Waas, M. *et al.* Droplet-based proteomics reveals CD36 as a marker for progenitors in mammary basal epithelium. *Cell Reports Methods* **4**, 100741 (2024).
32. Carabaña, C. *et al.* Spatially distinct epithelial and mesenchymal cell subsets along progressive lineage restriction in the branching embryonic mammary gland. *The EMBO Journal* **43**, 2308–2336 (2024).
33. Alečković, M. *et al.* Breast cancer prevention by short-term inhibition of TGF β signaling. *Nat Commun* **13**, 7558 (2022).
34. Burdziak, C. *et al.* Epigenetic plasticity cooperates with cell-cell interactions to direct pancreatic tumorigenesis. *Science* **380**, eadd5327 (2023).
35. Alonso-Curbelo, D. *et al.* A gene–environment-induced epigenetic program initiates tumorigenesis. *Nature* **590**, 642–648 (2021).
36. Baran-Gale, J., Chandra, T. & Kirschner, K. Experimental design for single-cell RNA sequencing. *Brief Funct Genomics* **17**, 233–239 (2017).
37. Kumar, T. *et al.* A spatially resolved single-cell genomic atlas of the adult human breast. *Nature* **620**, 181–191 (2023).
38. Yan, P. *et al.* Midkine as a driver of age-related changes and increase in mammary tumorigenesis. *Cancer Cell* **42**, 1936-1954.e9 (2024).

Point by point response to reviewer comments.

Original comments are in blue, authors replies are in black.
Changes are highlighted in green in the revised manuscript.

Reviewer #1 (Remarks to the Author):

The authors have responded to each of the points raised by the reviewers and included additional experiments to strengthen the study. The revised manuscript is improved, but still there are some remaining concerns.

We thank the reviewer for their thoughtful feedback, which has helped us further strengthen the manuscript.

The single cell RNA-seq analyses are still suboptimal, there are much more sophisticated data analysis and presentation tools than UMAP and bar plots.

We appreciate this comment and note that we have applied several sophisticated and state-of-the-art bioinformatics tools to analyze both newly generated and publicly available scRNA-seq data in this work. These include pseudotime trajectory inference methods such as PAGA and Slingshot, as well as the recently developed potency prediction tool, CytoTRACE 2 (PMID: 38562882), which leverages interpretable deep learning and is in press at *Nature Methods* (A. Newman, personal communication). Additionally, by employing UMAPs and bar plots (as well as violin plots and dot plots) for summarizing and visualizing complex scRNA-seq data, we followed best practice in the field, as exemplified by guidelines and display/analysis options in Scanpy and Seurat. However, we are happy to implement specific analyses if the reviewer has suggestions.

Many bar plots lack statistical tests and p-values (like Fig 3 g,h), also it's better to indicate actual p-values instead of asterisks for significance.

We thank the reviewer for raising this point – we have included the statistics in the **Fig. 3g, h, i** and have indicated p-values on all figures instead of asterisks for significance. For **Fig. 3g, h, i**, we used the Chi square test as the we have represented the fraction of IL33+/KRT6A+ cells over the total cells in the dataset. We had to use this method as some of the published datasets deposited in GEO do not have data on replicate samples. Moreover, we have performed immunofluorescence staining in independent samples for KRT6A (see **Fig. 4a, b**). We show that there is an accumulation of KRT6A+ cells in the aged nulliparous mice, confirming the single cell findings in **Fig. 3g** at the protein level.

The in vivo experiment with IL33 (new Figure 6) is useful, but it raises several new issues:

(1) IL33 acts on many other cell types, not just epithelial cells, so are the observed changes through IL33 directly acting on epithelial cells or changing the microenvironment?

We thank the reviewer for raising this important point. Indeed, we do see an increase in the stromal population and increase in ST2+ immune cells with IL33 treatment (**Extended Data Figure 9c, d**). It is entirely plausible that these IL33-dependent changes within the microenvironment can in turn impact on the epithelial populations. However, our *in vitro* studies

(Fig. 5) with mammary epithelial cells demonstrates that IL33 increases colony formation and induces the proliferation and expression of KRT6A in basal epithelial cells. Moreover, we see an increase in EdU+ cells in basal organoids treated with IL33, further suggesting that IL33 is directly affecting basal cell proliferation (Fig. R1). Therefore, our data support the notion that the effect of IL33 seen *in vivo* is due to its direct action on epithelial cells. Nevertheless, we cannot rule out additional cross talk between the microenvironment and epithelial cells *in vivo*. We have now added this part to our discussion (line 387-389).

Figure R1: IL33 treatment directly enhances proliferation in basal organoids, but not luminal organoids. (a) Representative immunofluorescence images of basal organoids treated with (top panel) or without (bottom panel) IL33 (10ng/mL). Organoids were generated from either luminal or basal cells sorted from 8-week-old nulliparous mice. Organoids were cultured for 9 days before the addition of EdU (10uM). After a 24hr chase, organoids were harvested. (b). Quantification of EdU+ cells in basal and luminal organoids exposed to IL33. Data represent the total number of EdU+ nuclei per total nuclei in a field of view (FOV). Statistical significance was determined by performing multiple unpaired t-test with Holm-Šídák correction.

(2) The authors only treated 3 mice and only for 5 days – this is too few mice to ensure reproducibility and significant and they need an explanation for how only 5 days of treatment has such dramatic effects on mammary ductal morphology. Was this due to proliferation or apoptosis? The authors should test this.

We thank the reviewer for raising this point. We have now expanded our analysis to include additional mice (n= 6, see revised Fig. 6). We see the same consistent changes in ductal morphology at 1 day post treatment (1dpt). To determine whether IL33 treatment induces proliferation or apoptosis *in vivo*, we performed EdU and Annexin V analysis (Revised Fig. 6, Extended Data Fig. 10, and Fig. R2). We find that IL33 treatment induces proliferation of both luminal and basal cells at 1 dpt, but proliferation is normalized at 8dpt (Fig. 6h, Fig. R2f, g).

At 1dpt and 8dpt there are small but significant differences in apoptosis in the basal compartment (Extended Data Fig. 10). Specifically, we observed a decreased percentage of cells in early apoptosis (DAPI-/AnnexinV+) in basal cells at both 1 and 8dpt. While insignificant at 1dpt (p = 0.06), we also observed a slightly elevated live population (DAPI-/AnnexinV-) in IL33-treated mice. These results could suggest a pro-survival program enacted by IL33 in basal cells. Despite observing significant differences in cell death within the luminal population between IL33- and PBS-treated mice, the differences were inconsistent and often conflicted between timepoints.

Given the relatively minor changes in apoptosis, we speculate that the effects of IL33 are largely due to changes in proliferation and not cell death.

Frankly, we were also surprised by the dramatic effects of IL33 on the mammary gland in such a short time. We therefore performed an additional timepoint where we treated mice with IL33 or PBS for only 2 days, administered a single dose of EdU on the second day, and analyzed on the third day (**Fig. R2a**, short exposure or “S.E.”). We find that even short treatment with IL33 induces significant changes in ductal morphology (**Fig. R2b-e**) and proliferation of mammary epithelial cells (**Fig. R2f-g**), consistent with the 1dpt cohort.

Figure R2: Two day-exposure to IL33 is sufficient to induce altered ductal morphology and epithelial proliferation. (a) Schematic of experimental approach. Mice in the short exposure (S.E.) cohort were treated with IL33 for two consecutive days, pulsed with EdU on day 2, and sacrificed for analysis at day 3. (b) Representative images of Carmine-Alum stained mammary gland wholemounts across timepoints and treatment cohorts. (c) Quantification of width (μm) of the main epithelial duct in PBS- and rIL-33-injected mammary glands from S.E., 1dpt and 8dpt timepoints ($n = 3$ mice/condition, 5-10 measurements per mouse). (d) Representative images of PBS- and rIL-33-injected H&E stains from S.E., 1dpt and 8dpt timepoints. (e) Lumen area ($\mu\text{m}^2 \times 10^3$) in H&E stained sections of PBS- and rIL-33-injected mammary glands from S.E., 1dpt and 8dpt timepoints ($n = 3$ mice/condition, 20 measurements per mouse). (f) Representative flow plots depicting EdU+ gates for basal (left two columns) and luminal (right two columns) cells across treatment groups (PBS and IL33) and timepoints (S.E., 1dpt, and 8dpt). (g) Quantification of EdU+ basal (top) and luminal (bottom) cells across treatment groups (PBS and IL33) and timepoints (S.E., 1dpt, and 8dpt). Statistical significance was determined by performing 2-way ANOVA with Šídák's multiple comparisons test.

While these results are surprising and demonstrate a rapid response to IL33, we speculate that these observations reflect IL33's function as an "alarmin" to promote proliferation of epithelial cells initiated by cellular stress or damage, as observed in Wu *et al.*, 2021 ([PMID: 34343135](https://pubmed.ncbi.nlm.nih.gov/34343135/)). Indeed, Wu *et al.* found that, as basal epithelial stem cells activate expression of *Il33* to proliferate in response to lung damage, *Krt6a* promoter activity is enhanced—an observation that was not recapitulated in *Il33*^{-/-} mice. These findings are in line with our work and illustrate an important, functional relationship that ties IL33, epithelial proliferation, and *Krt6a* expression together. Our future work will focus on understanding how IL33 induces these changes in the mammary gland epithelium.

(3) similarly, in the scRNAseq data (Fig 7 1-b) it would be important to plot the Ki67+ cells, since the authors claim that IL33 induces aberrant proliferation.

We have included UMAPs visualizing *MKI67* and *TOP2A* expression (Fig. R3a and Extended Data Fig. 11b). Interestingly, we do find that Cluster 5 has a high proportion of *MKI67*⁺ cells and *TOP2A*⁺ cells relative to all other clusters. However, it is important to note that in our aged nulliparous mice the hybrid cluster has no differences in cell cycle stage when compared with other epithelial clusters (Extended Data Fig. 5b).

Figure R3: Expression of proliferation-related genes in IL33+ human breast epithelial cells. (a) UMAPs showing expression of *IL33*, *MKI67*, and *TOP2A* in the HBCA (Kumar et. al) dataset.

Reviewer #2 (Remarks to the Author):

I thank the authors for addressing the questions raised. The manuscript is much improved.

We are delighted to hear the reviewers kind support and welcome their continued feedback.

Two minor comments:

1) The authors should use the integrated HBCA (Reed et al 2024 Nat. Genetics) to assess the expression of IL33. Its a much larger dataset which integrated the 7 main human scRNAseq studies (2.1million cells). They should also show if these cells are represented more or less in any of the datasets.

We thank the reviewer for suggesting this dataset. We have now analyzed 889,281 epithelial cells within the iHBCA dataset (**Fig. R4a**) (Reed et al., E-MTAB-13664). Overall, we find that there is a population of hybrid cells that is enriched in expression of *IL33* (**Fig. R4b**). Specifically, this distinct hybrid cluster is enriched in the *Kumar et al.* dataset that expresses both basal and luminal markers (**Fig. R4b**, black dashed circle).

Figure R4: Analysis of iHBCA. (a) UMAP of iHBCA annotated by dataset (left) and cell type (right). (b) Expression of *KRT18*, *KRT5*, *ELF5* and *IL33* in iHBCA. *IL33*+ cluster is highlighted with a dashed circle. (c) Percentage of *IL33*+ cells across all datasets (see also Table 1). (d) Percentage of *IL33*+ cells in each dataset split by age (see also Table 1). (e) Percentage of *IL33*+ cells in the *Kumar et al.* dataset split by age and parity. Please note that age at first pregnancy, BRCA1/2 status are not available in the metadata. Aged > 50 years, Mid < 50 years but > 35 years, Young < 35 years.

Moreover, in the Twigger dataset, there is also a population of *IL33*+ cells that only expresses luminal markers (**Fig. R4b**, pink dashed circle). As the Twigger dataset was generated using cells isolated from human milk, we speculate that the expression of *IL33* could reflect unique biology associated with lactation and immunity. Although a clear cluster wasn't visible in the other datasets, we were able to identify *IL33*+ cells in all of them (**Fig. R4c**). Therefore, we explored whether the percentage of *IL33*+ epithelial cells and/or expression of *IL33* changes with age in

the iHBCA. Overall, we find that the percentage of *IL33+* cells increases with age in all datasets except the Twigger dataset (**Fig. R4d**).

There are several important caveats to note regarding our analysis with human samples. Our manuscript explores the impact of early first pregnancies on aging. The available human single cell RNA-sequencing datasets do not include the age at first pregnancy (**Table 1**), which can have an impact on the hybrid cell population. In the only dataset that includes age at first pregnancy (Reed *et al.*), there is only one patient with an early pregnancy that does not have mutations in *BRCA1/2* (**Table 1**). Therefore, we cannot assess the combined effects of early pregnancy and aging in any of the datasets in a rigorous manner. Nevertheless, we used the Kumar *et al.* dataset to understand whether the percentage of *IL33+* cells changes with aging and pregnancy as this dataset has the greatest number of samples in the aged (>50 years of age) group (**Table 1**), although we do not have information of age at first pregnancy or *BRCA1/2* status. We find that there is a significant increase in the percentage of *IL33+* cells with age (p value = 0.03), but there is no significant difference in the aged nulliparous and parous samples (**Fig. R4e**).

Table 1. Metadata for the seven datasets within iHBCA.

Dataset	Fresh/ Frozen	# of cells	% of IL33+	Age at first pregnancy	# of parous patients >50 with no reported mutations in BRCA1/2	# of nulliparous patients >50 with no reported mutations in BRCA1/2
Kumar	Fresh	230298	1.49%	unknown	12	9
Pal	Fresh	75239	2.6%	unknown	0	2
Gray	Fresh	30677	1.5%	unknown	5	1
Reed	Frozen	343561	0.9%	known	6	1
Nee	Frozen	108184	0.4%	unknown	2	0
Murrow	Frozen	53071	0.2%	unknown	0	0
Twigger	Frozen	48251	9.7%	unknown	3	0

Importantly, we do not yet understand what other factors may affect the accumulation of this hybrid population. These could include race, ethnicity, BMI and mutation in *BRCA1/2*. Indeed, our analysis presented in **Extended Data Figure 11d** demonstrates an over-representation of *IL33+* hybrid MECs in Asian and African American donors. Preliminary analysis of *BRCA1* mutant breast tumors also shows an increase in the *IL33+* cells (data not shown). These factors/variables also influence breast cancer risk (American Cancer Society. *Breast Cancer Facts & Figures 2024-2025*. Atlanta: American Cancer Society; 2024). It is therefore possible that our findings may not translate into humans within certain contexts and/or populations.

Finally, although the iHBCA has generated an immensely useful resource for the community, four of the seven datasets in the iHBCA used frozen cells for sequencing (**Table 1**). Moreover, there were diverse methods used for tissue digestion, cell prep and sequencing (e.g., Reed *et al.* dataset used frozen, partially digested tissues). This can significantly affect the representation of rare cell types in the population. We have now included a paragraph highlighting the limitations of our analysis in human samples in the discussion section (lines 369-383).

2) The discussion starts with "Our study resolves and comprehensively maps the complex relationship between aging and pregnancy in the mammary gland, such as basal-cell bias and differentiation." I don't believe the current study fully resolves or comprehensively maps this relationship. I think this needs to be toned down to reflect the data presented.

We sincerely appreciate the reviewer's suggestions. We have now modified the sentence to read, "Our study lays the groundwork for understanding the complex relationship between aging and pregnancy in the mammary gland, such as basal-cell bias and differentiation". We have also revised the discussion section to more accurately reflect the findings, scope, and limitations of our study.

Reviewer #3 (Remarks to the Author):

The authors have been responsive to the critiques and this revision has improved many aspects of the previous data presented. Figs 1-5 are excellent and present a great coherent story.

We thank the reviewer for their enthusiasm towards our study and welcome their thoughtful feedback.

In response to reviewers asking for *in vivo* confirmation of IL33 effect, and for human validation, they added figures 6 and 7. However, these two new figures caused more concerns than solving problems.

IL-33-induced changes in mammary glands in mice at day 1 post completion of 5 days of treatment are not consistent with the cell culture and organoid data presented before this new figure 6 - KRT6A expression was induced in luminal cells but not basal cells but the reverse was observed in the *in vitro* experiments. More convincing explanations should be provided for this inconsistency.

We thank the reviewer for raising this important point. We acknowledge that the data from aged mice and *in vitro* data do not completely translate to our *in vivo* experiments in young mice. We hypothesize that this difference is because mammary basal cells *in vivo* are largely non-proliferative and do not give rise to luminal cells under homeostatic conditions (PMID: 26511661, PMID: 30271991, PMID: 21983963). However, in the organoid culture the basal cells are proliferative and differentiate towards luminal cells. Therefore, when the transition from basal to luminal *in vitro* occurs in the presence of IL33, it induces expression of *Krt6a*. We have now added this to our discussion so that readers have a nuanced understanding of our findings (lines 387-403).

Furthermore, to fully understand how IL33 regulates the differentiation of mammary epithelial cells in young and old mice in a biologically relevant manner will require several lines of experimentation that include the generation of complex mouse models (lineage-specific deletion and over-expression of *Il33*, and studies with young and 18-month-old mice). Due to these reasons, we feel that fully resolving the role of IL33 in mammary epithelial cells remains an important subject for future work.

Nevertheless, through multiple lines of evidence, our study establishes a population of IL33+ cells in the aged mammary gland that is reduced by early pregnancy and demonstrates that IL33 can induce proliferation and expression of KRT6A both *in vitro* and *in vivo*. This uncovers a previously

unknown role for IL33 in MECs and opens a new area of investigation into IL33 in mammary epithelial biology. We have included the clarification and limitations/future directions in the discussion section of the paper.

Furthermore, the authors did not detect an impact of IL-33 treatment at day 8, calling into question about the critical role of IL-33 in mediating the ageing effect on expanding the hybrid cell population. The authors should explain why the lack of detecting a more long-term change did not weaken their central conclusions.

We understand the reviewer's concern about the transient nature of IL33-induced phenotypes *in vivo*. We theorize that the natural aging process in aged, nulliparous mammary glands involves a more gradual, sustained exposure to IL33 in comparison to the short 5-day exposure in our *in vivo* model. Thus, in our model the effects of IL33 are resolved 8 days post-treatment. Moreover, we believe that the age-related accumulation and aberrant differentiation cannot be fully recapitulated with short-term IL33 treatment *in vivo*. To recapitulate these phenotypes will likely require the creation of a transgenic mouse line where IL33 is continuously expressed using a basal-specific promoter (Krt5 or Krt14), which is beyond the scope of the current work.

However, this does not affect our central conclusions, which are **1.** KRT6A/IL33+ cells accumulate in aged nulliparous mice, **2.** KRT6A/IL33+ cells are reduced in aged parous mice **3.** IL33 can induce proliferation and aberrant differentiation in mammary epithelial cells. We have revised the manuscript to provide a more comprehensive discussion of these findings, their associated limitations, and how they can be interpreted (lines 387-403).

*“While our in vitro data confirms the ability of IL33 to directly act on basal epithelial organoids, we cannot rule out the potential cross talk with the microenvironment, such as immune cells or fibroblasts that can impact MECs. We also find that IL33 induces KRT6A expression in basal cells in vitro as seen in the aged nulliparous mammary gland, while in young mice IL33 induces expression of KRT6A in luminal cells in vivo. We speculate that this difference is because the basal cells in vivo are largely non-proliferative and do not give rise to luminal cells under homeostatic conditions. However, in the organoid culture the basal cells are proliferative and differentiate to give rise to luminal cells. Therefore, when the transition from basal to luminal in vitro occurs in the presence of IL33, it induces expression of Krt6a. Notably, when basal cells acquire hybrid signatures due to expression of *Pik3ca*^{H1047R} or ablation of the luminal layer, they dramatically increase the expression of *Il3356*. We theorize that the natural aging process in aged, nulliparous mammary glands involves a more gradual, sustained exposure to IL33 in comparison to our short 5-day exposure in vivo model. Thus, in our model the effects of IL33 are resolved 8 days post-treatment. While our study can partially recapitulate some aspects of this phenotype in young mice with IL33 treatment, the age-related accumulation and aberrant differentiation cannot be fully recapitulated with short-term IL33 treatment in vivo. Further studies will be necessary to fully resolve the role of IL33 in mammary epithelial cells with aging.”*

Human scRNA data did not detect a hybrid cell population that is consistent with their mouse studies, and they did not detect any impact of ageing or parity on this cell population, calling into the question of whether their analysis was appropriate or proper datasets were used. In the absence of more convincing data, this figure probably should be dropped.

We thank the reviewer for raising this point. Indeed, currently available scRNA-seq datasets are inappropriate to rigorously perform the required analysis. We have outlined specific reasons in our response to Reviewer #2 above, which include a lack of information on age at first pregnancy, differences in BMI, race, ethnicity, and germline variants, as well as technical differences in performing the scRNA-seq analysis.

However, we show that the IL33+ hybrid population identified in our initial human scRNA-seq analyses data is enriched in several key genes in the mouse hybrid cells (**Fig. 7c, Extended Data Fig. 11b**). Therefore, we hypothesize that IL33 marks a subpopulation of human mammary epithelial cells with hybrid characteristics. Importantly, our *in vitro* experiments using primary human mammary epithelial cells show that IL33 enhances organoid formation and promotes a basal-cell state, accompanied by increased expression of the cancer stem cell marker CD44. In response to the reviewer's concern, we have included a dedicated paragraph in the discussion highlighting the limitations of our human data analysis. We trust that this will adequately address the reviewer's comment, but we are willing to remove the figure if needed (lines 369-383).

In conclusion, our study represents a first step to uncover the intricacies of the aging mammary gland and how it is impacted by early pregnancy — a knowledge gap that is of high interest to the scientific community and the public.

Point by point response to reviewer comments.

Original comments are in blue, authors replies are in black.
Changes are highlighted in green in the revised manuscript.

Reviewer #1 (Remarks to the Author):

The authors have responded to each of the remaining comments of the reviewers by revising the manuscript, including adding some additional experiments.

However, the dramatic increase in duct size after just 1 day of IL33 administration is not resolved conclusively (Fig 1b, f). The authors checked proliferation and apoptosis and found that IL33 mainly increases proliferation. However, to see such a dramatic (at least 2-fold) increase in duct size in 24 hrs would mean that virtually all cells in the duct double in 24 hrs – which is not matching their EdU labeling data showing that 10-20% of the basal and luminal cells have incorporated EdU in ~6 days. There is clearly an increase in proliferation, but the magnitude and time course of this does not match the dramatic phenotypic change that occurs in a day. Have the authors considered alternative explanations like swelling of the ducts – checked cell size and potential fluid accumulation?

We thank the reviewer for their suggestions. We want to clarify that the short exposure (S.E.) treatment group included 48hrs of exposure to IL33, not 24hrs as stated by R1. We suspect that R1 may have confused the EdU chase timeline (24hrs) with the IL33 exposure timeline. In any case, we respectfully disagree with R1's assertion that "*such a dramatic (at least 2-fold) increase in duct size in 24 hrs would mean that virtually all cells in the duct double in 24 hrs*". While we do observe a ~2-fold increase in width of primary ducts in IL33-treated mice from the S.E. group, we also observed a >18-fold increase in EdU+ luminal cells (0.4% vs. 7.7% EdU+) and nearly 10-fold increase in EdU+ basal cells (0.24% vs. 3.0% EdU+). Given the magnitude of these effect sizes, it is entirely plausible that enhanced proliferation is a critical driver of the IL33-induced ductal dilation phenotype.

While we do not see any fluid in the ducts with IL33 treatment by H&E staining, we cannot exclude that there is edema occurring in the glands. Unfortunately, we cannot fully resolve this as there are no commonly used techniques to accurately measure fluid in the ducts.

We also checked for changes in cell size by using forward scatter area as a proxy for cell size (**Figure R1**). While there are statistically significant differences in the FSC-A between treated and untreated mice, the directions of change (i.e., increase or decrease in FSC-A) between treatment groups do not correlate with the ductal dilation phenotype, which is only observed in S.E. and 1dpt time points (**Fig. R1a, b**). If IL33 is causing the ductal dilation phenotype through increasing cell size, we would expect increased FSC-A values in both S.E. and 1dpt cohorts. However, this prediction does not align with our data. In fact, basal and luminal cells from IL33-treated mice display a slight but statistically lower median FSC-A value relative to untreated mice at 1dpt (**Fig. R1b**). To account for differences in proliferation-related cell size changes, we excluded EdU+ MECs and repeated the same analysis. We observed no significant differences

in median FSC-A values in the 1dpt group, which is the group that demonstrated the most robust morphological phenotype (**Fig. R1c**).

To further interrogate whether altered cell size contributes to the IL33-dependent ductal dilation phenotype, we re-analyzed the flow data using a gating strategy developed to compare cell size in MECs based on SSC-A and FSC-A parameters (PMID: 23408103). Although we did find a slight increase in the proportion of large basal and luminal cells within the S.E. cohort, we did not observe a similar increase at 1dpt - the cohort which shows the maximum ductal dilation phenotype (**Fig. R2, Table 1**). These findings implicate that cell size has a minor contribution to the phenotype, if any.

Figure R2. Comparisons of small, medium, and large populations between treatments groups and across treatment regimens. (a) Representative gating strategy (left) and quantification (right) for small, medium, and large basal populations between treatments groups and across treatment regimens. (b) Representative gating strategy (left) and quantification (right) for small, medium, and large luminal populations between treatments groups and across treatment regimens. Gating strategy was adopted from Machado et al., 2013 (PMID: 23408103). Statistical significance was determined by performing two-way ANOVA with Tukey's multiple comparisons (Table 1).

Table 1. Statistical tests relating to Figure R2.

Tukey's multiple comparisons test	Mean diff.	95.00% CI of diff.	Adjusted P Value
BASAL			
Small			
SE -IL33 vs. SE +IL33	0.9333	-2.495 to 4.362	0.96
1dpt -IL33 vs. 1dpt +IL33	-1.807	-5.235 to 1.622	0.61
8dpt -IL33 vs. 8dpt +IL33	-5.833	-9.262 to -2.405	<0.001
Medium			
SE -IL33 vs. SE +IL33	10.2	6.771 to 13.63	<0.001
1dpt -IL33 vs. 1dpt +IL33	-1.867	-5.295 to 1.562	0.58
8dpt -IL33 vs. 8dpt +IL33	-3.167	-6.595 to 0.2619	0.08
Large			
SE -IL33 vs. SE +IL33	-11.23	-14.66 to -7.805	<0.001
1dpt -IL33 vs. 1dpt +IL33	4.133	0.7048 to 7.562	0.01
8dpt -IL33 vs. 8dpt +IL33	9.467	6.038 to 12.90	<0.001
LUMINAL			
Small			
SE -IL33 vs. SE +IL33	-2.833	-6.491 to 0.8245	0.21
1dpt -IL33 vs. 1dpt +IL33	-3.89	-7.548 to -0.2321	0.03
8dpt -IL33 vs. 8dpt +IL33	-5.797	-9.455 to -2.139	<0.001
Medium			
SE -IL33 vs. SE +IL33	10.73	7.075 to 14.39	<0.001
1dpt -IL33 vs. 1dpt +IL33	2.167	-1.491 to 5.825	0.49
8dpt -IL33 vs. 8dpt +IL33	-3.733	-7.391 to -0.07545	0.04
Large			
SE -IL33 vs. SE +IL33	-6.4	-10.06 to -2.742	<0.001
1dpt -IL33 vs. 1dpt +IL33	4.533	0.8755 to 8.191	0.008
8dpt -IL33 vs. 8dpt +IL33	9.333	5.675 to 12.99	<0.001

We hypothesize that the dilated duct phenotype is due to an imbalance between rapid proliferation, differentiation, change in basal to luminal ratio and reduced apoptosis induced by IL33. A similar phenomenon has previously been observed with FGF3 overexpression (PMID: 12169667) where induction of FGF3 expression leads to ductal dilation and an imbalance between proliferation and apoptosis. The FGF3-induced ductal dilation is reversible as well, similar to the phenotype that we see with IL33. Previous work from the Guo lab (PMID: 27653681) has also shown that mammary epithelial cells transduced with constitutively active PI3KCA and transplanted *in vivo* demonstrate a similar phenotype of dilated ducts which progresses to overt tumors in 9 months. This is interesting because the expression of mutant PI3KCA^{H1047R} in luminal cells upregulates IL33 when cells transition from luminal to basal fates (PMID: 26266985). We have included these points in the results and discussion of our manuscript.

As mentioned in our previous revision, fully resolving the role of IL33 in the mammary gland will require complex mouse models and is beyond the scope of the current work. We are very excited about our findings, and our future work is aimed at understanding the role of IL33 in proliferation and differentiation within the normal mammary gland, aging and tumorigenesis. We have also acknowledged this as a future direction in the discussion of our manuscript.

Reviewer #2 (Remarks to the Author):

Thank you for addressing my concerns. The analysis of the iHBCA presented in the rebuttal is informative and should be part of the main paper.

I have no further comments.

We thank the reviewer for reviewing our manuscript and as per the recommendation from the reviewer we have included the iHBCA analysis in the manuscript (**Extended Data Fig. 12**).

Reviewer #3 (Remarks to the Author):

The authors have been responsive to the reviewer's comments. The revised manuscript now include additional data and revised presentation to address the inconsistency between in vitro finding and in vivo validation and bioinformatic studies of published human scRNAseq data. Overall, the manuscript has strengthened, but the inconsistency moderates the significance of the study. Additional concerns are detailed below:

We thank the reviewer for their careful evaluation of our manuscript and acknowledge the minor typographical error (see point #1) that occurred on our end. We would like to emphasize that our conclusions are strongly supported by multiple independent lines of evidence demonstrating that the IL33+ population increases with aging, is reduced in aged parous mice, and that IL33 can partially recapitulate age-associated phenotypes. We believe that all concerns have now been fully addressed.

1. Fig 2b: the proportion plot shows the 18M P group has a lower proportion of all cell types. If the same total cell numbers were analyzed in 18M P vs 18M NP, reduction of some cell groups should be accompanied by decrease of others. In this study, nearly twice as many cells were used in the P group compared to NP group (NP, red, 3,327 cells vs P, blue, 7,305 cells), to this reviewer, it is mathematically impossible that the 18M P has lower counts for all cell subsets. As a result, all the statements regarding this fig are a bit shaky. To help readers, the authors should ensure that the same number of cells were used for this comparison in Fig 2A and 2B. Also since 3 samples are used, intra-sample variations should be shown.

We thank the reviewer for bringing up this point and apologize for the inadvertent typographical error on our part. The cell numbers reported in **Fig. 2b** legend for 18M NP corresponded to a subcluster rather than the total number of cells. We appreciate the reviewers attention to detail and have now corrected the cell numbers in the legend of **Fig. 2b**. As per the reviewer's request, we randomly downsampled the two groups to include the same number of cells (updated **Fig. 2b** and **Fig. R3**). The resulting proportions and trends are consistent with our original findings, confirming that the conclusions are not affected by differences in total cell numbers.

Figure R3. Bar plot showing the proportion of cell types by condition (*left* – all cells, *right* – equal numbers of cells in P and NP samples).

Regarding intrasample variability, this concern was previously addressed in our response to Reviewer #2 during Revision 1, as outlined below (in red).

Reviewer #2, Revision #1: Fig 3, It would be interesting to see the distribution of hybrid cells across samples. Are they primarily identified in specific samples, or are they generally upregulated in all aged NP samples?

*We appreciate the reviewer for raising this concern. To address this, we have compared the distribution of hybrid cells in our dataset with the 10x Tabula Muris Senis dataset (**Extended Data Fig. 4d**). Interestingly, we find that this population is significantly higher in the 10x Tabula Muris Senis dataset compared to ours – Hybrid cells in 10x Tabula muris – 78 cells, 18M NP – 13 cells, 18M P- 3 cells (**Extended Data Fig. 4d**). In our initial clustering, we found 27 cells in the 18M NP mice in our dataset but with reclustering to identify pericytes as requested in Point #6, some of these cells clustered with the basal cluster (**Extended Data Fig. 6b**). Importantly, we have now validated our findings using two independent aging datasets (**Fig. 3a-f**). We find that the number of cells in hybrid cluster is different across different experiments, but the population can be consistently identified in multiple datasets.*

Fig. R2: UMAP plot of scRNA-seq data generated from the mammary glands annotated by cell-types (left). Cells previously annotated as hybrid are displayed in red (right).

Of note, our single-cell RNA sequencing was performed on three biological replicates using hashing to distinguish individual samples. However, due to technical issues, the hashing procedure did not function as expected, preventing us from deconvoluting the replicates.

2. Fig 2a/b: The integration with Tabula Muris (red, n = 2,993 cells) is confusing. If more cells were indeed needed, the authors should sequence more cells from their own experiments. Integrated analysis should be secondary not primary, and readers should be convinced that the public data by others was comparable.

We thank the reviewer for raising this point. Our use of the Tabula Muris Senis dataset is based because it's Dr. Sikandar's previously published work as part of the Tabula Muris Consortium — During her post-doctoral fellowship, Dr. Sikandar was directly involved in sample preparation and annotation of mammary gland data in both Tabula Muris and Tabula Muris Senis. We also received the 18M Parous sequencing data before the 18M Nulliparous samples and therefore initiated our analysis using Tabula Muris Senis 10x dataset.

We are unable to repeat sequencing for the 18-month NP and P mice, as these aged cohorts are not available. Since identifying the hybrid population, we have focused our efforts on experimental validation, which we believe provides stronger biological insight. We have confirmed our findings at a protein level in an independent cohort of mice (**Fig. 4a, b**) as well as other datasets including mice with mutations in ESR1 (**Fig. 3**). Additionally, we have now performed immunofluorescence analysis across multiple ages, revealing a gradual increase in KRT6A+ cells with age. This **new** temporal analysis is included below (**Fig. R4**).

[FIGURE REDACTED]

Figure R4. Quantification of KRT6A+ MECs with aging in nulliparous mice. Statistical significance was determined by performing one-way ANOVA with Holm-Sidak correction. n = 2-3 mice per group, 3-5 images per mouse.

3. Fig 4a/b: this figure is used to show KRT6A+ cells are switched from the luminal layer to the basal cell layer with ageing. While the 18M NP panel shows convincing basal location of KRT6a+ cells, the 3M NP panel does not show convincing luminal cell staining for KRT6a.

We apologize for the image resolution in the manuscript. We have now provided new images for the panel. In general, KRT6A+ luminal cells are very rare at 3 months. Moreover, our data for 3M NP are consistent with published literature that KRT6A is expressed in luminal cells (PMID: 16790075, PMID: 21532625, PMID: 25019062).

4. Fig 5H: the organoid culture of Tp53 WT control basal cells does not show response to IL33. This seems to be inconsistent with Fig 5A.

We thank the reviewer for this comment. There is an increase in the control basal cells transduced with the pSICOR lentiviral plasmid but when we use a 2-way ANOVA with a correction for multiple hypothesis testing, we do not see a statistically significant difference. However, if we run a paired t-test to account for biological and technical variability in mice/different day/sorting/etc., significance is 0.05 (Fig. R5). This detail is now highlighted in the Figure 5 legend of the manuscript.

Figure R5. Paired t-test of GFP+ organoid count in pSICO-transduced basal organoids with and without IL33 treatment.

We hypothesize that basal cells, after undergoing enzymatic dissociation, sorting, and lentiviral transduction at low cell numbers (1000 cells), fail to proliferate and instead adopt a senescent-like state making them less responsive to IL33. In contrast, cells transduced with *shTrp53* gain a proliferative advantage. A previous study that successfully used basal cells for lentiviral screens expanded these cells in culture prior to transduction (PMID: 27653681). However, because basal cells spontaneously differentiate into luminal cells *in vitro*, such expansion would confound our analysis. As this figure is not essential to the main message of the manuscript, we are willing to remove it to avoid any confusion.

Our data conclusively demonstrates that IL33 increases basal organoid formation in mice and primary human mammary epithelial cells (HMECs) (**Fig. 5a and Figure 7d**). Moreover, we also see an increase in EdU+ cells in basal cells treated with IL33 as shown in our previous rebuttal letter (previously Figure R1).

To further support our conclusions, we have now examined whether loss of IL33 impacts proliferation of basal or luminal cells. To perform this experiment, we sorted basal and luminal cells from either wildtype or *Il33^{fl/fl}-GFP* mice (Jax Strain #:030619) and transduced with a lentivirus expressing Cre. Consistent with IL33 treatment, we find the loss of *Il33* through lentiviral delivery of the Cre recombinase construct significantly reduces basal but not luminal organoid formation (**Fig. R6**). Our future work using *Il33^{fl/fl}-GFP* mice will examine the role of IL33 in mammary gland development and aging.

[FIGURE REDACTED]

Figure R6. Loss of *Il33* reduces basal organoid formation. (a). Representative fluorescence and brightfield images of WT (*Il33*^{+/+}) and *Il33* KO (*Il33*^{fl/fl}) basal organoids infected with LES2A-Cre-mScarlet lentivirus. Scale bar reflects 2mm. (b). Quantification of basal organoid count in basal and luminal, WT and *Il33* KO organoids. A threshold area cutoff was set at 400 arbitrary units. (c). Validation of *Il33* KO by immunofluorescence staining. Scale bar reflects 200µm. Statistical significance was determined by performing multiple unpaired t test. n = 4 mice per group.

5. Line 262-264: The authors state that “Wholemount analysis of the mammary glands from IL33-treated mice showed a significantly increased lymph node area and dramatic dilation of the ducts at 1dpt (Fig. 6b, c).” But the phrase “increase lymph node area” is confusing. Do the authors mean increased ducts around the lymph node area? If so, an explanation is needed for this difference around the LN area compared to other parts of the mammary glands.

We apologize for the confusion - we meant the area of the lymph node itself that has increased and remains high at 8dpt. We have now revised the sentence in the manuscript.

6. Line 266-268: “Aside from LN area, analysis of mice at 8dpt showed no significant changes in any of the parameters above, suggesting the effects of short term IL33 treatment were transient (Fig. 6b-f).” This sentence does not make sense. Do the author mean changes are seen around the LN area, but not other parts of glands? If so, explanations are again needed.

We apologize for the confusion - we meant the area of the lymph node itself that has increased and remains high at 8dpt. We have now revised the sentence in the manuscript.

7. Fig 6I and J: IL33 treatment for 1 day (but now 8 days) led to a change of the ratio of basal vs. luminal cells. But this potentially important observation is not stated in the results section nor discussed.

We thank the reviewer for raising this point. We did highlight this difference in the flow cytometry analysis (line 273-274). We have also included it for the immunofluorescence staining (**Fig. 6i, j**) and have now noted in the manuscript that “*In addition, IL33 treatment in vivo increased the percentage of SMA+ cells (Fig. 6i, j) consistent with our flow cytometry analysis (Fig. 6h), suggesting the IL33 treatment can partially recapitulate some age-related phenotypes in vivo.*”

Point by point response to reviewer comments.

Original comments are in blue, authors replies are in black.

Reviewer #1 (Remarks to the Author):

The authors response to the remaining comment is fine. Could they also add the statement that expansion of the ducts due to swelling cannot be excluded to the manuscript also when they are describing the data? the enlargement of the ducts is a very dramatic phenotype (even if it was 48 hrs not 24 hrs treatment) so I'm sure others will also question this.

We thank the reviewer for the feedback. As requested, we have included the statement, "*H&E staining confirmed that there was a significant increase in size of the lumen in IL33 group at 1dpt (Fig. 6e, f) but did not reveal fluid accumulation within the ducts; however, we cannot entirely exclude the possibility of dilation secondary to edema*" to the manuscript (lines 267-269).

We have also included the cell size analysis as part of Extended Data Fig. 9c-e and in the manuscript (lines 287-290).

The responses to reviewer 3 and revision seem OK to me, although I do not see the figures to make sure resolution is publishable quality- but this you can check during QC.

We thank the reviewer for their comments. We have now uploaded high resolution Illustrator files that are publishable quality.